# A nanotrap improves survival in severe sepsis by attenuating hyperinflammation

Changying Shi[1,7], Xiaojing Wang[1,7], Lili Wang[1,7], Qinghe Meng[2], Dandan Guo[1], Li Chen[3], Matthew Dai[1,6], Guirong Wang [2,4], Robert Cooney[2,4] & Juntao Luo [1,2,4,5✉]

Targeting single mediators has failed to reduce the mortality of sepsis. We developed a telodendrimer (TD) nanotrap (NT) to capture various biomolecules via multivalent, hybrid and synergistic interactions. Here, we report that the immobilization of TD-NTs in size-exclusive hydrogel resins simultaneously adsorbs septic molecules, e.g. lipopolysaccharides (LPS), cytokines and damage- or pathogen-associated molecular patterns (DAMPs/PAMPs) from blood with high efficiency (92–99%). Distinct surface charges displayed on the majority of pro-inflammatory cytokines (negative) and anti-inflammatory cytokines (positive) allow for the selective capture via TD NTs with different charge moieties. The efficacy of NT therapies in murine sepsis is both time-dependent and charge-dependent. The combination of the optimized NT therapy with a moderate antibiotic treatment results in a 100% survival in severe septic mice by controlling both infection and hyperinflammation, whereas survival are only 50–60% with the individual therapies. Cytokine analysis, inflammatory gene activation and tissue histopathology strongly support the survival benefits of treatments.

[1] Department of Pharmacology, State University of New York Upstate Medical University, Syracuse, NY 13210, USA. [2] Department of Surgery, State University of New York Upstate Medical University, Syracuse, NY 13210, USA. [3] Department of Pathology, Baylor Scott and White Medical Center, Temple, TX 76508, USA. [4] Sepsis Interdisciplinary Research Center, State University of New York Upstate Medical University, Syracuse, NY 13210, USA. [5] Upstate Cancer Center, State University of New York Upstate Medical University, Syracuse, NY 13210, USA. [6] Present address: Brown University, Providence, RI 02912, USA. [7] These authors contributed equally: Changying Shi, Xiaojing Wang, Lili Wang. ✉email: luoj@upstate.edu

Sepsis is a life-threatening condition caused by an exaggerated inflammatory response of the host to systemic infection. Heterogeneous features of the host and bacterial interactions create a complex, dynamic, and nonlinear disease that is extremely challenging to treat[1,2]. The mortality rates in patients with severe sepsis and septic shock remain 30–41%, despite advanced supportive care[3]. The current treatment for sepsis includes antibiotic administration to control the septic foci, fluid resuscitation, and vasopressors as needed to maintain organ perfusion. During sepsis, the release of PAMPs and DAMPs perpetuates systemic inflammation, hemodynamic instability, and organ failure[4,5]. Unfortunately, effective therapies to prevent overwhelming inflammation and its sequelae in sepsis are lacking.

Lipopolysaccharide (LPS, a potent PAMP) released from the outer membrane of Gram-negative (GN) bacteria binds to Toll-like receptors (e.g., TLR-4) on immune cells[6], stimulating the production of inflammatory cytokines[7], and triggering severe systemic inflammation in sepsis[3,8]. Therapies like anti-TLR-4 antibodies[9,10] and LPS-binding molecules/polymers[11–14], including polymyxin B (PMB, a LPS-binding antibiotic)[15,16], were leveraged to control LPS-related inflammation, but have yet to reduce mortality in patients with sepsis. Therapies developed to neutralize proinflammatory cytokines (e.g., TNF-α[17–19] and IL-1β[20,21]) also failed to improve sepsis mortality in clinical trials[22]. We believe that this is due to the biologic variability of inflammatory response in infected patients, and the fact that multiple inflammatory mediators contribute dynamically to systemic inflammation in sepsis. Based on this concept, we believe that therapies attenuating multiple mediators are promising interventions, which could be used in conjunction with antibiotics and supportive care to improve survival of sepsis.

We developed a well-defined linear–dendritic telodendrimer (TD) nanoplatform with the precise and engineerable chemical structures for customized nanocarrier design in drug delivery[23,24]. We introduced multiple charges and hydrophobic moieties on the dendritic periphery of TD for efficient protein encapsulation[25,26]. The charge and hydrophobic structures are ubiquitously present in inflammatory mediators, including cytokines, LPS, and DAMP/PAMP molecules[27], like cell-free DNA/RNA, extracellular adenosine triphosphate (ATP), and free heme[28]. The flexible TD scaffold freely changes its conformation to effectively bind a broad range of inflammatory mediators through a synergistic combination of multivalent electrostatic and hydrophobic interactions. Interestingly, we notice that most proinflammatory cytokines are negatively charged proteins, whereas most anti-inflammatory cytokines are positively charged. This charge disparity may poise the activity regulation by the acidosis during inflammation and sepsis[29,30]. In addition, cytokines and DAMPs/PAMPs generally have smaller molecular weights of <30 kDa than abundant serum proteins. Here, we report to rationally synthesize the TD NT in size-exclusive hydrogel resins, e.g., PEGA resin with pore size ~50 kDa[31], to preferably capture inflammatory mediators with different charges, respectively, for precise and effective immune modulation. Our study demonstrates a 100% survival of severe sepsis in murine models treated with the optimized TD NT in combination with a moderate antibiotic therapy by attenuating both hyperinflammation and infection.

## Results
**TD NT for LPS binding**. As illustrated in Fig. 1a, our previous studies have shown TD nanocarrier design for protein encapsulation and delivery based on the synergistic combination of multiple charge and hydrophobic interactions[25,26]. Similarly, the negatively charged LPS can be effectively captured by TDs composed of positively charged and hydrophobic moieties as illustrated in Fig. 1b. For nomenclature, TD was named, for example, $PEG^{5k}(ArgVE)_4$ to refer the PEG (5 kDa)-tethered oligolysine dendron terminated with four arginine (Arg) and four vitamin E (VE). Other acidic or basic amino acids or derivatives, e.g., lysine (Lys), aspartic acid (Asp), and glutamic acid (Glu) or oxalic acid (OA) can be conjugated on TD periphery together with hydrophobic moieties, e.g., heptadecanoic acid (C17), vitamin E (VE), and cholesterol (CHO), respectively, via standard peptide chemistry as shown in Supplementary Fig. 1 following our previous publications[25]. TD $PEG^{5k}(ArgVE)_4$ binds LPS and assembles into micelles with ~24 nm in size, which was smaller than individual TD (30 nm) and LPS (30 nm) nanoparticles (Supplementary Fig. 2). The fluorescent polarization (FP) spectrometry studies indicated the efficient complexation formation between FITC-LPS with TD nanotraps (NTs) than LPS–PMB complexation (Supplementary Fig. 3).

LPS isolated from different GN bacteria, e.g., *E. coli* and *P. aeruginosa*, can be efficiently loaded in $PEG^{5k}(ArgVE)_4$ TD nanoparticles (Fig. 1c), which are stable in the presence of 40-fold excess of PMB (Fig. 1d). In contrast, the PMB–LPS complex with a moderate micromolar-binding affinity[32,33] is less stable and dissociates in electrophoresis (Fig. 1d). As shown in Fig. 1e, the TD–LPS nanocomplex remains stable in the presence of serum protein, which is correlated with the stronger TD–LPS binding than TD–BSA interactions revealed in molecular docking studies (Supplementary Fig. 4). The supplement data in Supplementary Fig. 5 showed that TD nanocarriers can efficiently trap DAMP molecules, e.g., free DNA and small-molecule-free heme.

**Nanotrap (NT) immobilized on the size-exclusive hydrogel resins**. LPS and most cytokines have relatively small molecular weights (10–30 kDa). Therefore, we conjugated the TD nanotrap on size-exclusive hydrogel resins to selectively capture these inflammatory mediators and exclude the abundant large serum proteins, such as albumin and immunoglobulin (Fig. 2a). We applied standard solid-phase peptide chemistry to synthesize TD stepwise on three hydrogel resins with different pore sizes, e.g., commercial PEGA resin[31] and a synthesized PVA–PEG resin following our previous literature[34] (Supplementary Fig. 6). At the same time, TD was synthesized on a cleavable Rink resin following the same procedure for TD structure confirmation after cleavage. MALDI-TOF MS and NMR analysis confirmed the precise TD synthesis on the hydrogel resins (Supplementary Fig. 7). The nanotrap resins with the arginine and hydrophobic moieties (R) are denoted as $RESIN-(ArgR)_4$.

As expected, the combination of both electrostatic and hydrophobic interactions in TD nanotrap resin is important for effective LPS adsorption: 10–20% versus 95% for TD NT resins with individual moieties or combinational interactions (Supplementary Fig. 8A, B). LPS-adsorption capability of PEGA-TD NT resins can be regenerated with sustained adsorption efficiency after four cycles of regeneration, whereas the efficiency of PEGA-PMB was reduced gradually by 50% (Supplementary Fig. 8C). To test the diffusion kinetics of proteins with different molecular weights, we synthesized two Förster resonance energy transfer (FRET) peptides on PVA–PEG and PEGA resins, e.g., $Y(NO_2)$ dlHKSriK(Abz) and $Y(NO_2)$ayGrGrrK(Abz)[35], as substrates for Trypsin (24 kDa) and TNKase (Tenecteplase) (45 kDa), respectively. Upon enzymatic cleavage, the fluorescent quencher $Y(NO_2)$ (3-Nitrotyrosine) will be cleaved; Abz(ortho-aminobenzoic acid) moieties on hydrogel resins become fluorescent gradually (Supplementary Fig. 9). As shown in Fig. 2b, smaller trypsin diffused ~30-fold faster than TNKase in PEGA resin, which indicated excellent selectivity toward smaller proteins like cytokines over larger serum proteins. PEGA has a molecular

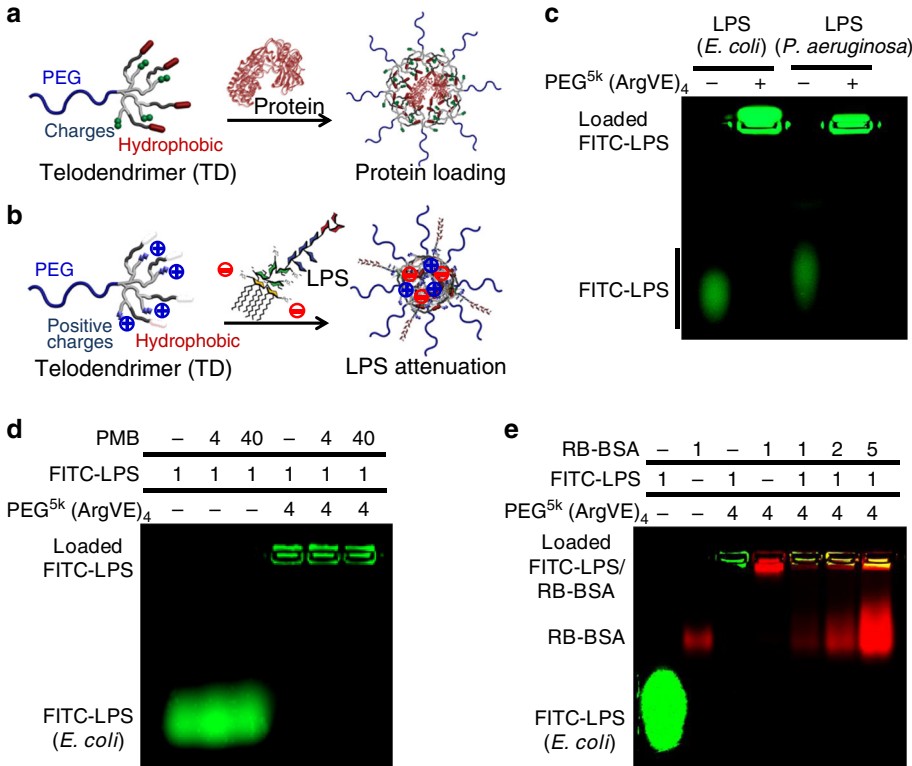

**Fig. 1 Protein and LPS binding in telodendrimer nanoparticles.** Schematic illustration of **a** protein and **b** LPS captured by telodendrimer nanoparticles via the combination of charge and hydrophobic interactions. **c**–**e** Agarose gel electrophoresis profiles reveal the complex formation of the FITC-labled LPS with telodendrimer PEG[5k](ArgVE)[4], as indicated by the lost mobility in migration (repeated independently a minimum of twice). **c** LPS originated from both *E. coli* and *P. aeruginosa* can be captured efficiently by the telodendrimer PEG[5k](ArgVE)[4]. **d** PMB form less-stable complex with LPS in electrophoresis, which was also unable to dissociate LPS– PEG[5k](ArgVE)[4] nanocomplex with 40-fold excess in mass ratio. **e** The stability of LPS–PEG[5k](ArgVE)[4] nanocomplex was also observed to be stable in the presence of serum protein (RB-BSA) at different mass ratios.

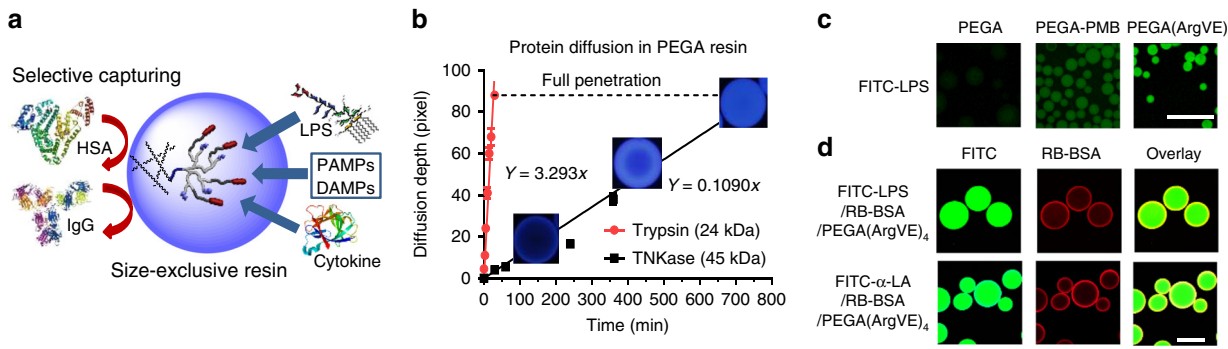

**Fig. 2 TD NT resins for selective adsorption based on size effects. a** Schematic representation of selective LPS and cytokine removal by nanotrap-immobilized size-exclusive resin. **b** Kinetic diffusion of proteases Trypsin (24 kDa) and TNKase (45 kDa) in PEGA resins conjugated with the corresponding substrates: beads become fluorescent upon substrate cleavage by enzymes releasing the quencher (nitrotyrosine) from the fluorescent dye (Abz) (see Supplementary Fig. 9). **c** Fluorescent microscopy images showing the adsorption of FITC-LPS by blank PEGA, PEGA-PMB, and PEGA-(ArgVE)[4] resins (scale bar: 500 μm). **d** Confocal images of PEGA-(ArgVE)[4] resins incubated with the mixture of FITC-LPS or FITC-α-LA with RB-BSA (1:100 mass ratio), indicating the effective penetration of smaller LPS and α-LA and restriction of larger BSA binding on the surface of resin (scale bar: 100 μm). **c**, **d** Repeated independently a minimum of twice.

weight cutoff ~50 kDa[31], and exhibits a better size-exclusive effect against large proteins than PEG–PVA resin (Supplementary Fig. 9); therefore, it was selected for further studies. PEGA-(ArgVE)[4] resin adsorbs FITC-LPS more efficiently than blank PEGA resin and PEGA-PMB resin, indicated by stronger fluorescence adsorbed on beads (Fig. 2c). Further, confocal images revealed that FITC-LPS and FITC-α-LA (14.2 kDa, PI: 4.5) diffused throughout PEGA-(ArgVE)[4] hydrogel resins without

interference by the presence of abundant RB-BSA (1:100 mass ratio), which was restricted to the surface adsorption (Fig. 2d).

**LPS attenuation from biological fluids.** Fetal bovine serum (FBS) and whole blood from healthy human volunteer were doped with FITC-LPS and were incubated with 10% volume of PEGA-(ArgC17)[4] resins and other resins, respectively, e.g., acetylated PEGA blank resin, PEGA-PMB and two commercial

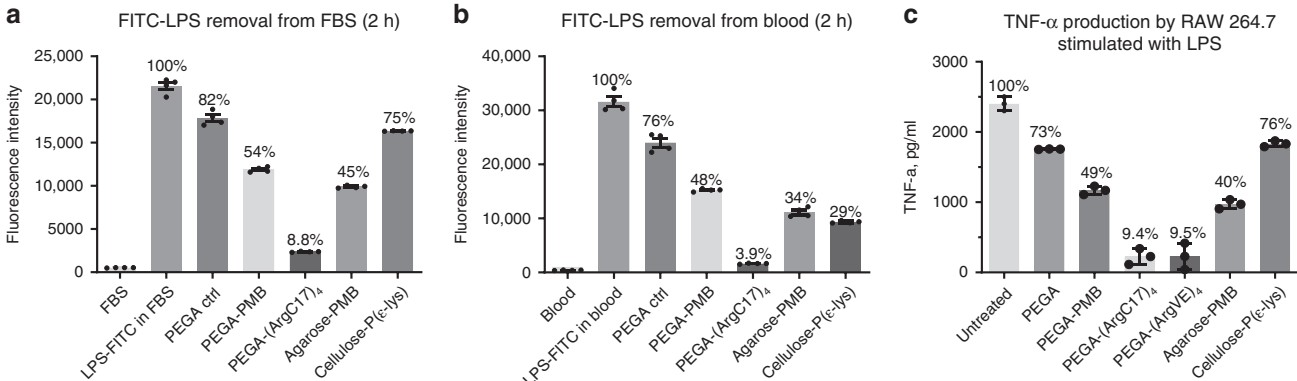

**Fig. 3 LPS attenuation by TD NT resins.** The removal of FITC-LPS (12.5 μg/mL) by nanotrap PEGA hydrogel resins in comparison with commercial LPS-removal resins (**a**) in FBS and in (**b**) whole blood after 2-h incubation, respectively. **c** The stock LPS solution was pretreated with or without different LPS-adsorption resins before adding to RAW 264.7 cells at an untreated LPS concentration of 500 ng per mL. TNF-α production in the culture medium after overnight incubation was analyzed by ELISA assay ($n = 4$, mean ± SEM). Source data are available in the Source Data file.

LPS-binding resins, agarose–PMB resin and cellulose–poly(ε) lysine. As shown in Fig. 3a, PEGA-(ArgC17)₄ removed FITC-LPS from FBS with ~91% efficiency; PMB-based resins only removed 46–55% of LPS, and polylysine resin was only slightly better than the PEGA control (25% vs. 18%) after 2-h incubation, which was close to the equilibrium as detected after overnight incubation (Supplementary Fig. 10). PEGA-(ArgC17)₄ also removed high levels of FITC-LPS from whole blood with an efficiency of ~96% after 2-h incubation (Fig. 3b). LPS elimination from blood by PEGA-PMB (52%), agarose–PMB (64%), and cellulose–polylysine resins (70%) was less effective after 2-h incubation. Accordingly, PEGA-(ArgC17)₄ resins showed higher intensity after above-FITC-LPS incubation than all other resins (Supplementary Fig. 11). Unlabeled LPS was added directly or after resin incubations into the culture medium of murine macrophage RAW 264.7 cells. Culture medium was collected after overnight incubation, and TNF-α level was analyzed by ELISA (Fig. 3c). PEGA-TD resins inhibited TNF-α production by >90%, which was significantly better than the PMB-containing PEGA resin (51%) and agarose resin (60%). Polylysine-modified cellulose resin exhibited similar TNF-α attenuation with the control resin by 25%, which was correlated with its poor LPS removal in FBS (Fig. 3a).

**Selective protein adsorption.** Since charge interactions are critical in protein binding by TD nanocarriers and TD nanotrap resins, we investigated the isoelectric points (PIs) of critical cytokines in the database[36]. Interestingly, we noticed significant charge disparity in counteracting cytokines in both human (Fig. 4a) and mouse (Supplementary Fig. 12): most proinflammatory cytokines (TNF-α, IL-1, IL-6, IL-12, and HMGB-1) have negative charges with PIs ranging between 4.1 and 6.4, while the most common anti-inflammatory cytokines (IL-10, TGF-β, IL-4, and IL-11) have positive charges (PIs: 8.2–11.7). We can simply change the charge groups in TD NT to target-specific group of cytokines for potential precise immune modulation. Two model proteins were selected to mimic proinflammatory and anti-inflammatory cytokines for adsorption test, e.g., α-lactalbumin (α-LA, 14.2 kDa, PI: 4.5) and lysozyme (Lyz, 14.4 kDa, PI: 10.7). We synthesized TD nanotrap (NT) moieties on PEGA resins with C17 fatty acid as hydrophobic moieties, which further were conjugated with either positively charged arginine (Arg) or negatively charged oxalic acid, yielding NT⁽⁺⁾ PEGA-(ArgC17)₄ and NT⁽⁻⁾ PEGA-(OAC17)₄, respectively. We then tested the charge selectivity in protein adsorption as shown in Fig. 4b. As expected, the attractive

electrostatic interactions significantly improve protein adsorption in NTs: for example, the negative protein α-LA⁽⁻⁾ was adsorbed much more efficiently in PEGA-(ArgC17)₄ NT⁽⁺⁾ than the positively charged lysozyme⁽⁺⁾ (~95% vs. 25% after 30-min incubation); vice versa, NT⁽⁻⁾ adsorbs lysozyme⁽⁺⁾ much more efficiently than trapping α-LA⁽⁻⁾. Interestingly, Arg-containing NT⁽⁺⁾ exhibited much faster adsorption kinetics than the OA-containing NT⁽⁻⁾ with the identical C17 moieties for both α-LA and lysozyme, which may be due to the capability of guanidinium for hydrogen bonding and pi-cation interactions with proteins. Both PEGA-(ArgC17)₄ and PEGA-(ArgVE)₄ resins with positive charges scavenged FITC-α-LA efficiently (94 and 96% after 2-h incubation, respectively, Fig. 4c), whereas, only 10% of α-LA was trapped physically in PEGA control resin. The PEGA-PMB resin also removed ~74% of negatively charged α-LA from FBS, owing to the positive charge and hydrophobic features of PMB (Fig. 4c).

A protein mixture of α-LA, Lyz, and BSA (1:1:10 in mass ratio) was incubated with PEGA-(ArgC17)₄ resin to study the kinetics and selectivity of protein adsorption. α-LA concentration significantly decreased compared with both larger BSA and positively charged Lyz after 30-min incubation, and was almost undetectable after 1-h incubation by MALDI-TOF MS (Fig. 4d). A neutral small protein myoglobin⁽⁰⁾ (Mb, PI: 7.1, 16.7 kDa) was added to the above protein mixture, then incubated with different charged nanotrap resins, respectively. As shown in Fig. 4e, blank PEGA resin did not alter the relative intensities of proteins compared with the stock mixture. The positively charged PEGA-(ArgC17)₄ resin adsorbed α-LA efficiently from the solution, and a noticeable decrease of neutral protein Mb was also observed. In contrast, the negatively charged PEGA-(OAC17)₄ resin completely scavenged the positively charged Lyz (PI: 11.6) with α-LA remaining in the solution. Mb was also removed more efficiently by NT⁽⁻⁾ PEGA-(OAC17)₄ than NT⁽⁺⁾ PEGA-(ArgC17)₄, maybe due to the chelation effects of oxalic acid (OA) with iron–heme complex in Mb. The absorbed α-LA and LYZ were eluted from resins with 8 M urea treatment, then analyzed by MALDI-TOF MS (Fig. 4f). Weak signals were observed in the elution from blank PEGA resin. In contrast, dominant signals of α-LA or Lyz were detected in the elution from the positive and negative nanotrap resins, respectively, confirming the efficient charge selectivity for protein adsorption. The loading capacity of α-LA in PEGA-(ArgC17)₄ resin was detected by MALDI-TOF MS analysis to be ~13 μg of α-LA per mg resin using BSA as an internal reference (Supplementary Fig. 13).

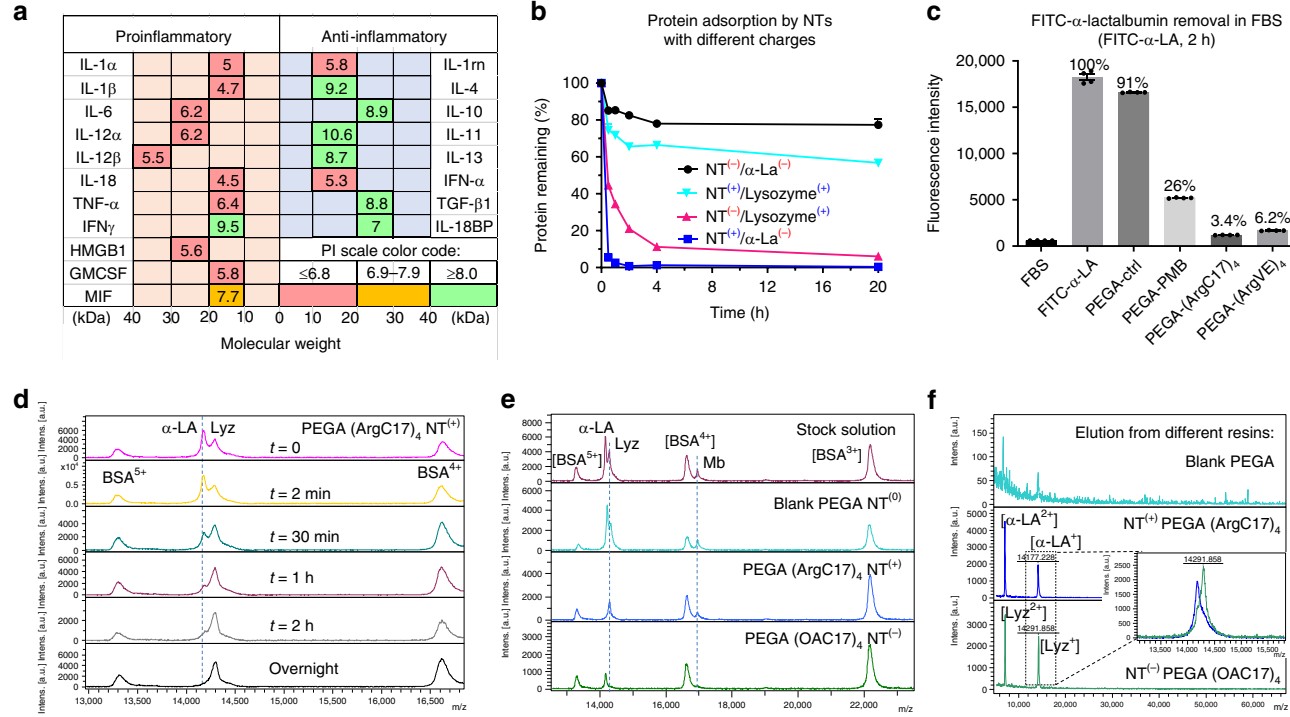

**Fig. 4 Selective protein adsorption in TD NT resins based on charge interactions. a** Summary of the molecular weights and isoelectric points (PIs) of key proinflammatory and anti-inflammatory cytokines in human sepsis. **b** The kinetic adsorption profiles of FITC-labeled $\alpha$-LA$^{(-)}$ and lysozyme$^{(+)}$ by nanotrap (NT) resins with positive (arginine) or negative (oxalic acid (OA)) charges, respectively. NT$^{(+)}$: PEGA-(ArgC17)$_4$; NT$^{(-)}$: PEGA-(OAC17)$_4$ ($n = 2$, mean ± SD). **c** The adsorption efficiency of negatively charged FITC-labeled $\alpha$-lactalbumin ($\alpha$-LA, 14.2 kDa, PI: 4.5) by various positively charged resins in FBS after 2-h incubation ($n = 4$, mean ± SEM). **d** MALDI-TOF MS analysis of the protein mixture solution of $\alpha$-LA (14.2 kDa, PI: 4.5, 0.5 mg per mL), lysozyme (Lyz, 14.4 kDa, PI: 10.7, 0.5 mg/mL), and BSA (66 kDa, PI: 4.2, 5 mg/mL) before and after incubation with NT$^{(+)}$ PEGA-(ArgC17)$_4$ resin for different time at a bead/solution ratio of 1:10 v/v. Selective $\alpha$-LA adsorption was observed. **e** MALDI-TOF MS analysis of the protein mixture solution of $\alpha$-LA$^{(-)}$ (0.1 mg per mL), lysozyme$^{(+)}$ (Lyz, 0.1 mg per mL), myoglobin$^{(0)}$ (Mb, 0.1 mg per mL, PI: 7.1, 16.7 kDa), and BSA$^{(-)}$ (1 mg per mL, PI: 4.8–5.4, 66.4 kDa) before and after incubation with blank-acetylated PEGA, positive NT$^{(+)}$ PEGA-(ArgC17)$_4$, and negative NT$^{(-)}$ PEGA-(OAC17)$_4$ resins, respectively, at bead/solution ratio of 1:4 v/v. Charge-specific protein adsorption was observed. **f** MALDI-TOF MS analysis of proteins eluted from nanotrap resins after protein adsorption with 8 M urea: weak signals were observed from blank resin eluent, and strong signal and charge selectivity were observed for charged TD resins. Source data are available in the Source Data file.

**Inflammatory modulation in sepsis treatment**. Sepsis models induced by cecum ligation and puncture (CLP) mimic the pathogenesis and progression of human sepsis[37]. To test cytokine adsorption effects of NT resins, septic mice were sacrificed 24 h after CLP (Fig. 5a), and plasma was collected for incubation for 2 h with NT$^{(+)}$ resin PEGA-(ArgC17)$_4$ at 10:1 plasma/resin volume ratio to mimic the duration for clinical hemoperfusion. As shown in Fig. 5b, NT$^{(+)}$ resin incubation significantly reduced the levels of IL-1β and IL-6 from septic plasma with 93.7–98.6% efficiency analyzed by ELISA. Early cytokine TNF-α peaks at 2–4 h in mice post CLP induction[38], and was detected at relatively low concentration in septic plasma at 24 h post CLP, which was even undetectable after NT$^{(+)}$ resin incubation. At the same time, NT$^{(+)}$ resin also decreased the positively charged IL-10 levels by 70% (Fig. 5b), which may also be helpful to prevent the development of immunosuppression.

Next, we directly applied blank PEGA resin and PEGA-(ArgC17)$_4$ NT$^{(+)}$ resin to the right side of abdominal cavity immediately after CLP to examine the effects of local immune intervention on mortality (Fig. 5c). As shown in Fig. 5d, septic mortality was 62.5% 48 h post CLP surgery in control group, and the blank PEGA resin did not alter CLP mortality. Surprisingly, the simultaneous peritoneal application of NT$^{(+)}$ resin actually increased CLP mortality rate to 77.8%. This finding confirms the activity of the NT$^{(+)}$ resin to target proinflammatory cytokines,

which however, may be harmful if the initiation of the innate immune response was inhibited in the presence of infections. Therefore, we examined the timing of NT resin application on survival in CLP sepsis, which is clinically relevant since sepsis patients generally have significant inflammatory symptoms at their first clinical appearance.

We applied NT resins with positive or negative charges in the peritoneal cavity in septic mice at different time after CLP (0 h, 3 h, and 8 h) to target the excessive proinflammatory or anti-inflammatory molecules, respectively. Again, the immediate application of NT$^{(+)}$ did not provide any survival benefit compared with CLP alone (Fig. 5e). However, concurrent application of NT$^{(-)}$ resin to target anti-inflammatory cytokines in CLP slightly increased survival rate to 37.5%. The delayed applications of both types of NT resins at 3 h and 8 h post CLP significantly improved the survival rates to 50–62.5% compared with CLP alone (25% survival). Interestingly, NT treatments at 8 h post CLP were better than 0-h and 3-h treatments, which was, however, not indicative of the later treatment of the better effect, since mice start to die 24 h post CLP. We further analyzed the blood cell counts over 42 days to monitor the status of the survived mice as shown in Fig. 5f. Survived mice in CLP control group had the lowest levels of the total white blood cells (WBCs) on day 2, followed by a dramatic elevation on day 14, indicating the continuous inflammation with hematological instability[39].

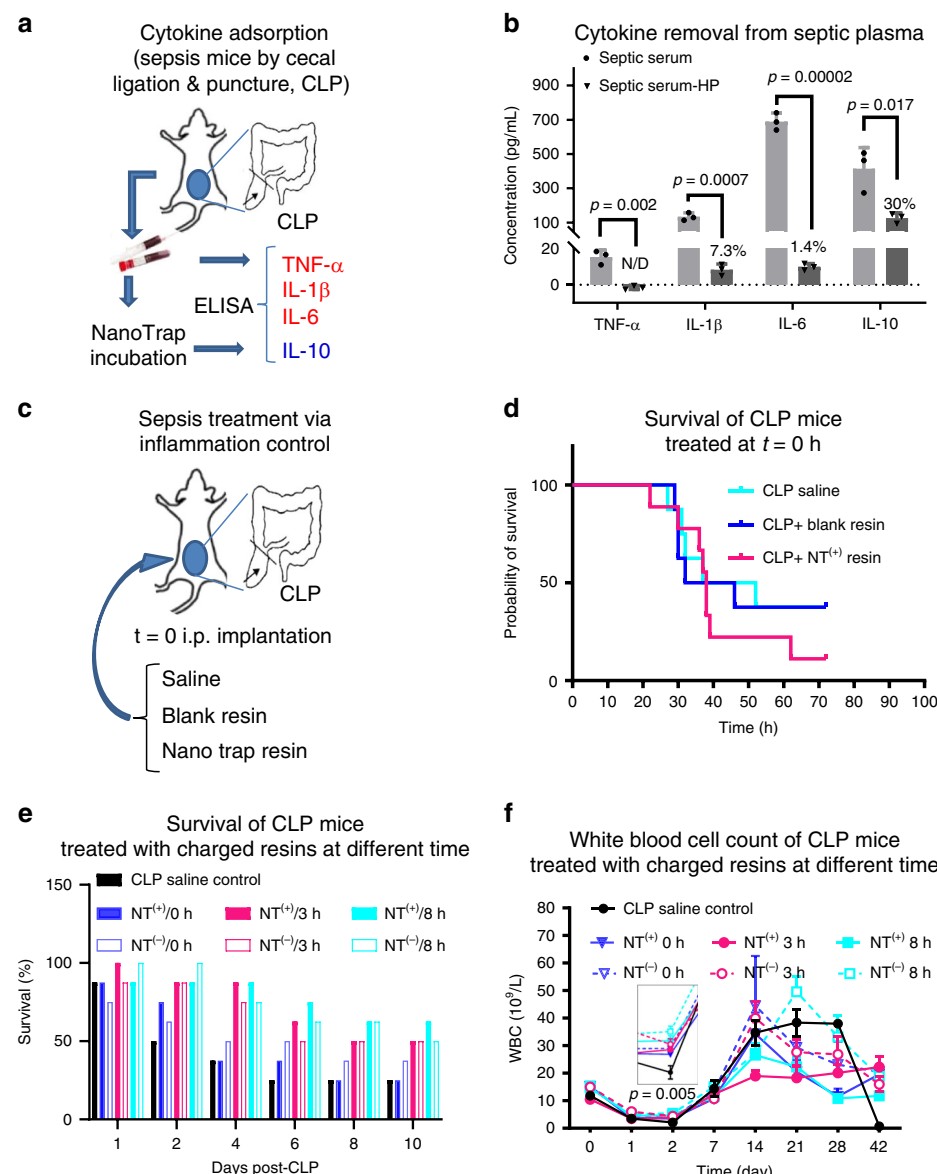

**Fig. 5 Immune modulation via TD NT resins. a** Schematic illustration of sepsis mouse model induced by cecal ligation and punctuation (CLP) procedure, and septic blood was collected 24 h post CLP for ex vivo bead incubation. **b** Key cytokines TNF-α, IL-1β, IL-6, and IL-10 in plasma were quantified via ELISA assays before and after 2-h incubation ($n = 3$, duplicated measurements, mean ± SEM. Statistical significance was measured by paired one-sided Student's test). **c** Experimental design of the spontaneous intraperitoneal treatments of CLP mice ($n = 8$) with saline, blank PEGA resin, or PEGA-(ArgC17)₄ NT⁽⁺⁾ resin in situ right after CLP procedure. **d** The survival of the animal was monitored for 3 days. Surprisingly, higher mortality was observed for spontaneous NT⁽⁺⁾ treatment, which may be due to the spread of the infection with the disabled innate immune response by effective resin attenuation. **e** The survival of CLP mice ($n = 8$) treated with intraperitoneal implantation of NT⁽⁻⁾ PEGA-(OAC17)₄, or NT⁽⁺⁾ PEGA-(ArgC17)₄ at 0 h, 3 h, or 8 h after CLP without antibiotics. **f** The white blood cells were monitored for the survived CLP mice in the above treatments over 42 days (statistical significance was measured by unpaired one-sided Student's test). Source data are available in the Source Data file.

Interestingly, optimal hematological profile was observed in mice treated with NT⁽⁺⁾ at 3 h post CLP, WBC returned back to a normal level on day 7, and remained stable thereafter. In contrast, other treatment groups exhibited much more hematological instability. Especially for NT⁽⁻⁾ resin treatments, the prolonged inflammation and chronic critical illness were evidenced by the formation of necrotic peritoneal abscesses[40]. Based on the above findings, the schedule for resin administration was fixed at 3 h post CLP in the following studies.

**Combination therapy in sepsis treatment**. The effective control of both infection and inflammation is critical to prevent multiple organ failure and improve survival in sepsis. Drug resistance is an

increasing challenge in the clinic for infection control. The hypoperfusion in sepsis[41] hinders the drug delivery to the infection sites, thereafter, compromising the antibiotic effect in sepsis. To mimic such clinical scenario, we applied a moderate dose of antibiotic imipenem/cilastatin (IMI, 50/50 mg/kg, 50% of its full dose in mice[42]) at 3 h after CLP to partially control infection in CLP mice. At the same time, we apply both NT⁽⁺⁾ and NT⁽⁻⁾ resins to modulate immune reactions through intraperitoneal implantation at 3 h post CLP with or without IMI administration, respectively.

The combination of NT⁽⁺⁾ and antibiotics (Fig. 6b) yielded a 100% survival before euthanasia on day 42, which was statistically significant compared with all other groups. Mice treated with

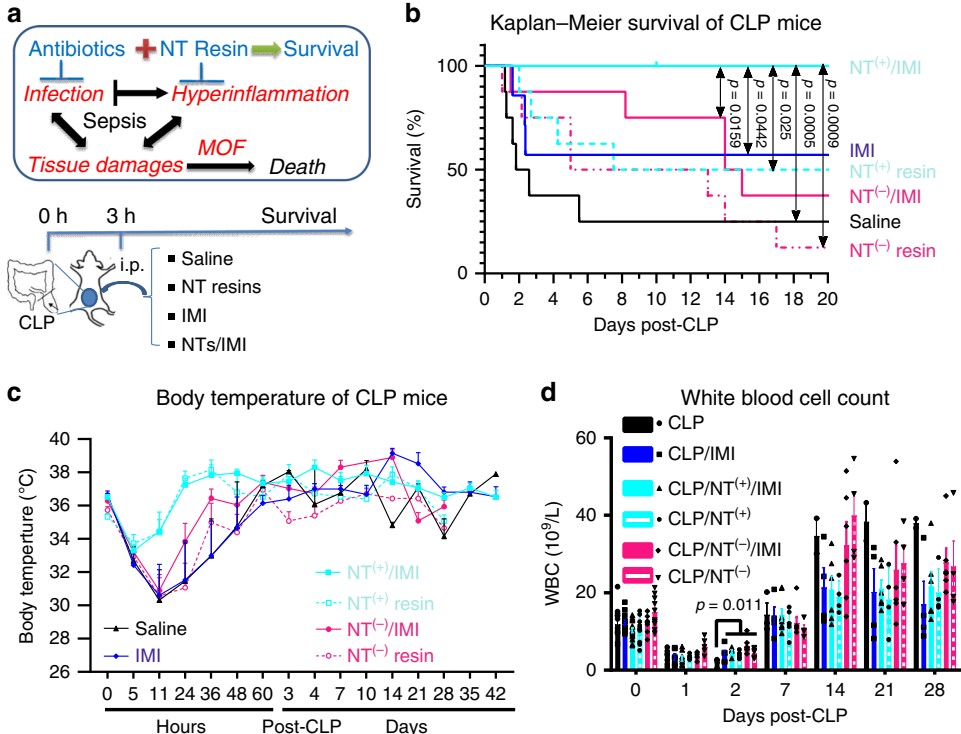

**Fig. 6 Sepsis treatment via TD NT resins. a** Schematic illustration of the mechanism of sepsis associated with multiple organ failure (MOF) and death driven by both infection and hyperinflammation and our therapeutic strategy by combining antibiotics and immune modulation via NT resin scavenging to control both infection and hyperinflammation through i.p. implantation on day 3 post-CLP in mouse sepsis models. **b** The survival of CLP mice ($n = 8$, except for IMI group, $n = 7$) treated with NT$^{(-)}$ PEGA-(OAC17)$_4$ or NT$^{(+)}$ PEGA-(ArgC17)$_4$ 3 h post CLP with or without antibiotics imipenem/cilastatin (IMI, 50/50 mg per kg body weight) treatments. As a result, the combination of IMI and NT$^{(+)}$ provides a 100% survival by treating both infection and hyperinflammation. After 1 week post CLP, animals in NT$^{(-)}$ and NT$^{(-)}$/antibiotic groups were continuously sacrificed upon severe necrotic abscess observed (statistical significance was analyzed by Log-rank (Mantel–Cox) test, and a significant level of 0.05 was used for comparison). **c** The reduced hypothermia and fast body-temperature recovery were observed in the groups treated with NT$^{(+)}$ w/wo IMI. **d** The white blood cells (WBC) were monitored for the survived CLP mice in the above treatments over 4 weeks ($n = 8$), and reduce over time as shown in **b**. Significant low WBC was observed in CLP control group on day 2, and stable white blood cell counts were observed in NT$^{(+)}$/IMI treatment groups with all mice survived (mean ± SEM, statistical significance was measured by unpaired one-sided Student's test, and a significant level of 0.05 was used for comparison). Source data are available in the Source Data file.

antibiotics alone had an ~50% mortality on day 7, which was lower than CLP-saline group (mortality >80% on day 7). Similar to the previous treatment studies in Fig. 5e, NT resins alone also yielded comparable survival benefit to the antibiotic treatment, and slower mortality incidences were observed mostly between days 2 and 6 post CLP, whereas mortality events in both CLP and antibiotic groups were concentrated at 24–72 h post CLP, likely due to hyperinflammatory reactions. The combination of NT$^{(-)}$ and antibiotics also significantly reduced the acute-phase mortality. However, more animals in this group were euthanized at the later stage upon severe abdominal abscess formation, indicative of prolonged illness. The body temperature (Fig. 6c) and white blood cell count (Fig. 6d) were recorded for the survived mice over time. The reduced hypothermia and fast temperature recovery were observed in the groups treated with NT$^{(+)}$ with or without antibiotics. CLP mice treated with NT$^{(+)}$ and antibiotics remained the most stable white blood cell counts over a long-term observation with all mice survived. In contrast, large amplitudes of hematological dynamics were observed in other groups, especially for NT$^{(-)}$ groups with/without antibiotics, correlating with the severe abscess formation (Fig. 6b).

The above animal studies were performed using female BALB/c mice in order to reduce the variability. Further, we tested the efficacy of the optimal NT$^{(+)}$/antibiotic therapy in male BALB/c mice ($n = 10$) with the same age (8–10 weeks) for CLP sepsis treatment in comparison with saline control and NT$^{(+)}$ or

antibiotic treatments (Supplementary Fig. 14). Interestingly, a slightly reduced mortality rate (~40% for the acute phase and overall 50% mortality over 6 weeks) was observed compared with female mice (60% on day 2, Fig. 6b) with the same CLP procedure. The reduced mortality might be due to the heavier body weight and more adipose tissue for male mice than the female at the same age. The different amount of peritoneal adipose tissue was reported to interfere significantly with the insults and immune response to CLP procedure[43]. Nevertheless, the combination of NT$^{(+)}$/antibiotics consistently yielded a 100% survival (Supplementary Fig. 14A) and stable hematological profile (Supplementary Fig. 14B) during 6-week observation. We further conducted other sepsis treatment studies in aged mice (11 months) induced by the same CLP procedure. The combination of NT$^{(+)}$/antibiotics again yielded a 100% survival, whereas 75 and 60% mortality rates were observed on day 7 in CLP mice treated with saline and IMI antibiotics, respectively (Supplementary Fig. 15).

**Reduced inflammation and tissue damages.** To improve sepsis treatment, it is important to demonstrate the evidence-based efficacy in severe sepsis. We conducted a separate batch of CLP studies ($n = 5$) with the identical procedure and treatments as shown in Fig. 6 with a sham laparotomy group included as a comparison. Body temperature of CLP mice reflects the severity

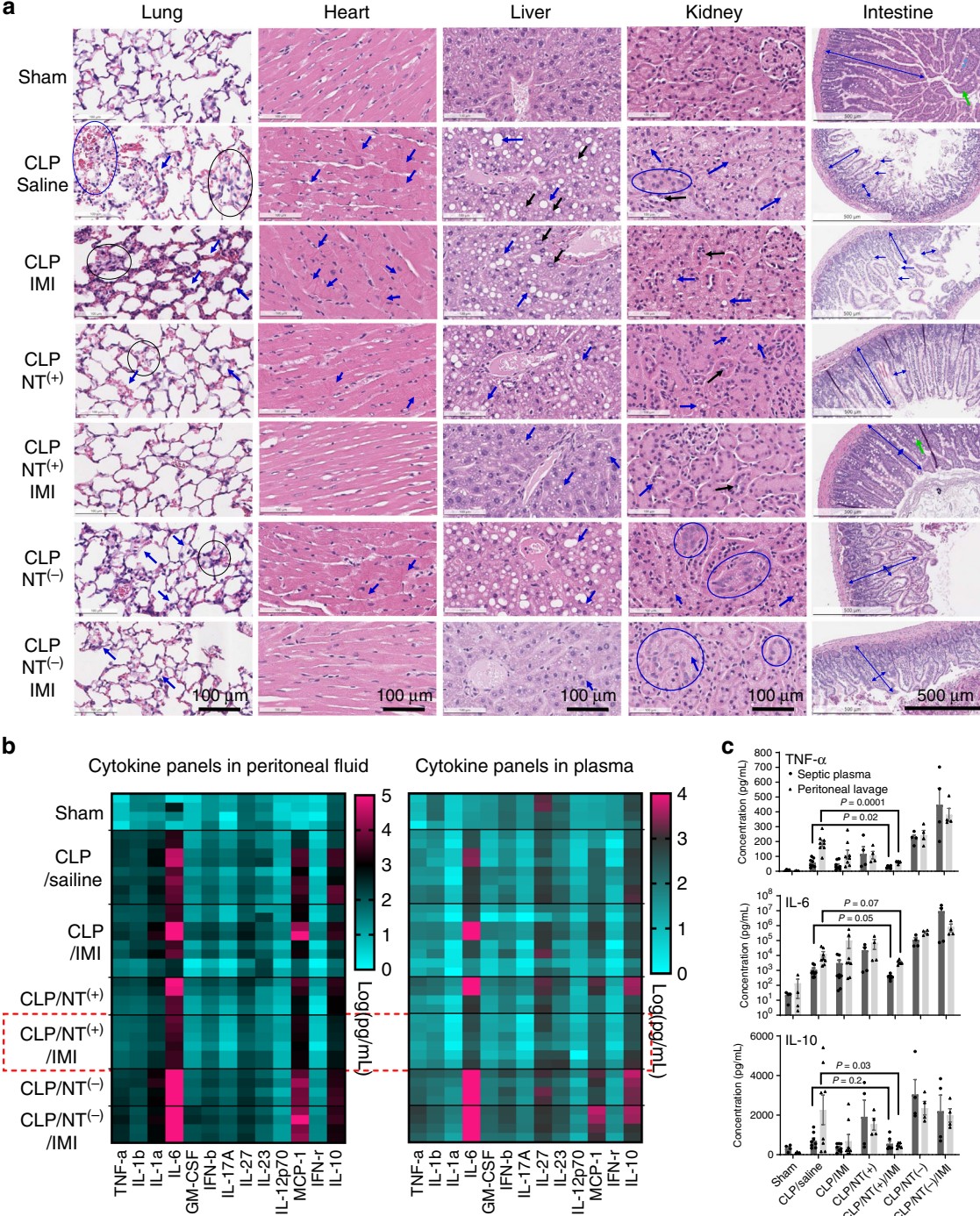

**Fig. 7 Reduced tissue damage and attenuated hyperinflammation in severe sepsis. a** Tissue histology ($n = 2$) stained with hematoxylin and eosin for major organs of septic mice with the most severe hypothermia at 24 h post CLP in different groups. Histopathological findings: Lung: alveolar congestion (black circle), hemorrhage (blue circle), alveolar thickening, and hyaline membrane formation (arrow); Heart: intracellular edema and contraction bands (arrow), indicating early cell death; Liver: predominantly macrovascular steatosis (blue arrow) and nuclear inclusion (black arrow); Kidney: proximal tubular edema and vacuolization (blue arrow), luminal epithelial desquamation (black arrow), and epithelial nuclear enlargement (circle); Intestine: villous shortening, villous edema, villous necrosis (blue arrow), and loss of goblet cells (green arrow). **b** The heatmap of multiple inflammatory cytokine profiles, and **c** quantitative analysis of key cytokines (IL-6, TNF-α, IL-1β, and MCP-1) in both plasma and peritoneal lavage fluids in mice at 24 h post CLP in different treatment groups in comparison with sham mice ($n = 4$, mean ± SEM. Statistical significance was measured by unpaired one-sided Student's test). Source data are available in the Source Data file.

of sepsis and correlates well with the mortality[44]. Hypothermic mice with body temperature <30 °C[44] were identified from each group (mostly 21–26 °C), and euthanized at 24 h post CLP for pathophysiological comparison. As shown in Fig. 7a, obvious

pathological changes and tissue damages were observed in multiple organs in severe septic mice. Significant features of the acute lung injury (ALI) were shown in CLP-saline mice, e.g., hemorrhage, alveolar thickening, alveoli congestion, and interstitial

edema. Concurrently, significant intracellular edema and contraction bands were observed in cardiomyocytes in CLP-saline mice (Fig. 7a), which is indicative of early cell death (necrosis) uniquely for cardiac myocytes[45]. Significant steatosis and vacuolization in the liver were observed in CLP mice, indicating the dysfunction and hepatocellular injury[27]. Significant focal vacuolization, epithelial cell flattening, and desquamation were observed with the resultant luminal dilation, indicating the tubular injury in the kidney. As shown in Fig. 7a, normal intestine has long intact villi with abundant goblet cells to maintain the protective mucus layer. In contrast, the CLP mice showed significant villous shortening, villous edema, villous necrosis, and loss of goblet cells in the intestine, which lead to the increased permeability of the epithelium barrier for microbiome dissemination.

It was noticed that the antibiotic IMI treatment alone did not improve histopathology in all organs as examined in Fig. 7a. Instead, enhanced inflammation and organ damages were observed in the IMI group, e.g., increased alveolar thickening and increased hepatocyte vacuolization, which may be due to the enhanced LPS release after bacteria killing[46]. NT$^{(+)}$ resin treatment improved the tissue damages to some extent by scavenging inflammatory mediators, for example, the reduced lung injury and less contract bands in cardiac myocytes. Further, the combination of IMI with NT$^{(+)}$ resins synergistically prevented the tissue damages in all vital organs. In contrast, significant tissue damages were still observed in mice treated with the negatively charged NT$^{(-)}$ resin, which were also improved by IMI combination. However, renal tubular injury and ALI remained significant in the NT$^{(-)}$/IMI group, indicating possible mortality.

As shown in Fig. 7b, heatmap of multiplex cytokine analysis revealed the significant increase of both proinflammatory and anti-inflammatory cytokines in peritoneal fluids in CLP mice in all treatment groups, e.g., TNF-α, IL-1α, IL-1β, IL-6, MCP-1, and IL-10. In plasma, the key inflammatory cytokines, e.g., IL-6 and IL-10, were also significantly elevated, indicating the spread of systemic inflammation. The individual treatments via antibiotic IMI or NT$^{(+)}$ only reduced the inflammation in individual animals. As expected, the combination of IMI/NT$^{(+)}$ significantly attenuates the inflammation both in peritoneal fluids and in blood (Fig. 7b), especially for TNF-α, IL-6, and IL-10 (Fig. 7c). In contrast, NT$^{(-)}$ treatment significantly increased inflammatory cytokines TNF-α, IL-6, and IL-10, which were even higher than CLP-saline control group. It may be explained by the attenuation of the positively charged anti-inflammatory cytokines by NT$^{(-)}$, which unleashes proinflammatory signaling and in turn leads to even higher IL-10 production.

Significant intestinal hyperemia was induced by CLP. The intestine was harvested at 24 h post surgery for cytokine analysis. As shown in Fig. 8a–c, significant increase in key proinflammatory cytokines, e.g., TNF-α, IL-1β, and IL-6, was observed in the intestine in CLP mice compared with the sham group. CLP mice treated with NT$^{(+)}$/IMI exhibited the lowest levels of intestinal cytokines among treatment groups and close to the sham group, especially for IL-6. HMGB-1 is a proinflammatory cytokine to propagate inflammation reactions through TLR-4 binding. It is also a typical DAMP molecule indicative of cell damage. As shown in Fig. 8d, the plasma levels of HMGB-1 were significantly increased in all CLP groups, indicating the systemic tissue damage. NT$^{(+)}$/IMI treatment had the lowest level of plasma HMGB-1 among all treatment groups. The similar trend for HMGB-1 was also observed in both intestine and liver (Fig. 8e, f), indicating the reduced organ inflammation and tissue damages by NT$^{(+)}$/IMI treatment. NF-κB activation is one of the most important pathway for inflammation in both immune system and

organs[47]. As shown in Fig. 8g, both NF-κB P65 and p-iκB-α levels in the liver of CLP mice treated with NT$^{(+)}$/IMI were as low as the sham animals, which were significantly upregulated in other groups (Fig. 8h, i), indicating the activation of NF-κB pathway. In summary, the histological and molecular analysis revealed that the synergistic combination of NT$^{(+)}$ and antibiotics attenuated the excessive inflammation and prevented organ damages, which support the survival benefit in severe sepsis.

## Discussion

Multiple signals and pathogenic pathways in sepsis are deemed to be targeted simultaneously in order to improve the survival of sepsis. The application of PMB failed in improving sepsis treatment, because of the dual functions of antibiotics and LPS attenuation. Accordingly, PMB-based Toraymyxin® hemoperfusion therapy also failed in the clinic trails for sepsis treatment[48]. The DAMPs and PAMPs perpetuate strong host inflammatory reactions, cytokine storm, and organ damages in sepsis. Our TD nanotrap possesses an "octopus-like" flexible dendritic scaffold, which maximizes the conformational entropy in binding with various inflammatory mediators via the ubiquitous and synergistic charge and hydrophobic interactions, e.g., LPS (Fig. 1), cytokines (Fig. 5), and DAMPs/PAMPs (Supplementary Fig. 5). Conventional adsorption resins are made of hydrophobic polymers for nonspecific adsorption of biomolecules, for example, Cytosorb® for multiple cytokine adsorption. However, the spectrum of molecular adsorption in these cartridges is fixed by the chemistry of the resin, which unfortunately was insufficient to improve the survival of sepsis in the clinical trials[49]. In contrast, we chose a hydrophilic, inert, and antifouling PEG-based PEGA resin for immobilization of versatile TD NT, which avoids cell adhesion (Supplementary Fig. 11). We further examined the biocompatibility of PEGA NT resins by intraperitoneal implantation in mice for 6 months. No noticeable acute or chronic toxicity was observed as evidenced by the normal body weight and blood counting and histological analysis (Supplementary Fig. 16).

The management of hyperinflammatory reactions is as important as effective infection control in bacteremia sepsis, which is even critical for viral sepsis, given no effective antiviral drugs. The precise immune modulation is critical for sepsis treatment because of the dynamic and dysregulated immune system in patients. The disparity of the surface charges in proinflammatory and anti-inflammatory cytokines (Fig. 4a) provides us a unique opportunity for the preferential cytokine attenuation. The charge disparity is conserved in human cytokines (Fig. 4a) and murine cytokines (Supplementary Fig. 12), which ensures the translation of the charge-based TD NT immune modulation from preclinical murine models into the clinical efficacy. Our studies indicated that the application of NT resins with different charges at different time after CLP resulted in different survival benefits in sepsis mice (Fig. 5d, e), emphasizing the importance of precise immune intervention.

We acknowledge that some limitations need to be addressed for clinical translation of TD NT approaches for sepsis treatment. First, we applied NT resin suspension directly in the abdominal cavity of CLP mice, which is not applicable for clinical use in sepsis patients, although it was biocompatible for long-term implantation (Supplementary Fig. 16D). Biodegradable and injectable hydrogels can be applied to tether TD nanotraps for local injection or topical application to attenuate pathogenic biomolecules. Alternatively, it is straightforward to pack TD NT resins into a cartridge for abdominal ultrafiltration or hemoperfusion therapy for clinical sepsis treatments. Second, NT$^{(+)}$ resins for hyperinflammation control will not be applicable to

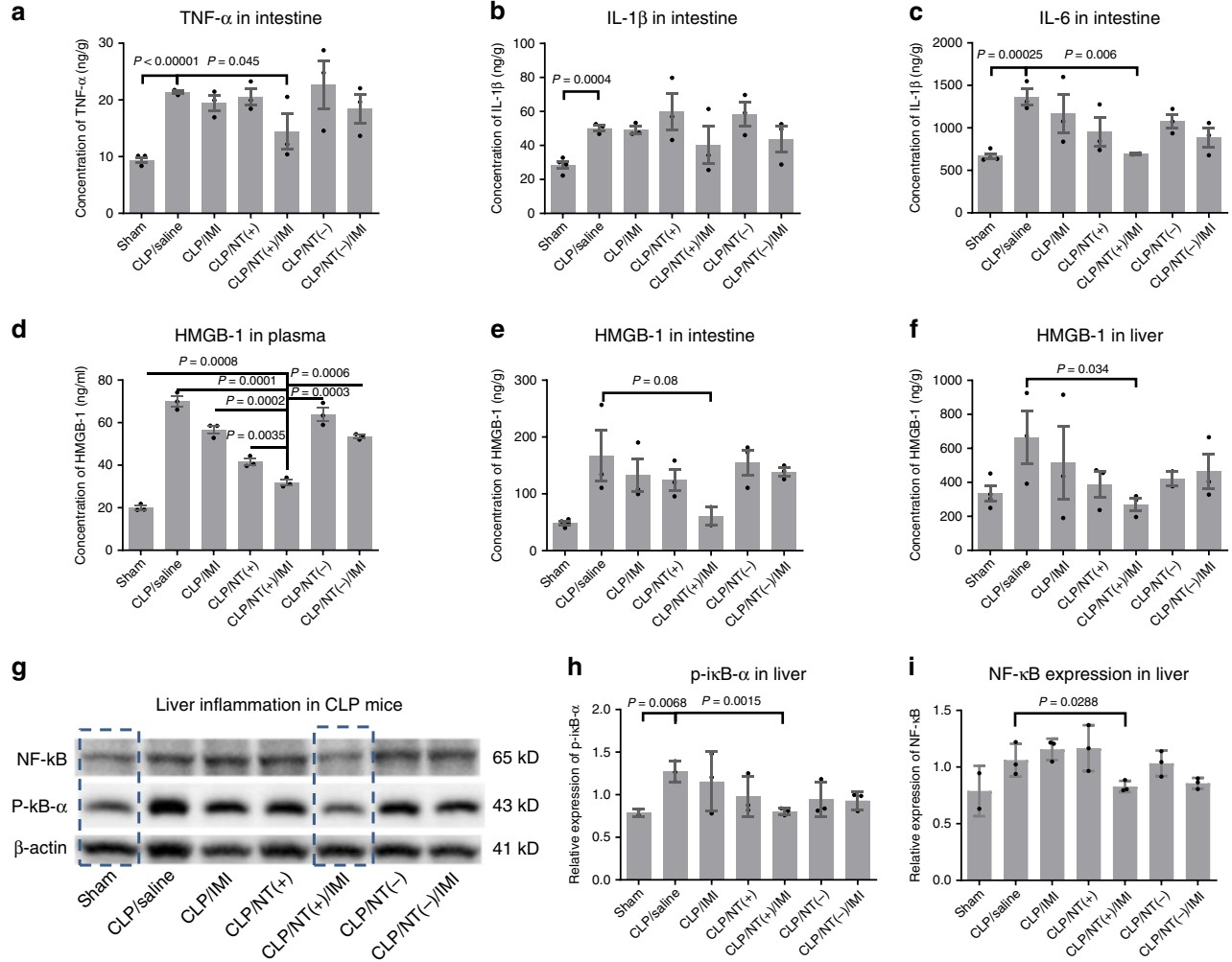

**Fig. 8 Reduced tissue damage and organ inflammation in sepsis survivor. a–c** Critical cytokine expression levels in the intestine. **d–f** HMGB-1 expression levels in the plasma, liver, and intestine as a DAMP indicator for tissue damage. **g–i** The expression levels of NF-κB and its activation via phosphorylation of IκB-α in the liver as an indicator of remote organ inflammation ($n = 3$, mean ± SEM. Statistical significance was measured by unpaired one-sided Student's test). Source data are available in the Source Data file.

sepsis patients in the later stage of immunosuppression. In this case, the application of NT$^{(-)}$ adsorption resins might be beneficial to harness the host immunity for infection control and tissue repair. In addition, this approach may not be sufficient to reverse the multiple organ failures, instead to prevent the progression of MOF by attenuating hyperinflammation.

In summary, we developed a versatile TD NT approach for effective immune modulation. The flexible multivalent charge and hydrophobic moieties on TD can be freely engineered for selective and efficient attenuation of a group of inflammatory mediators for sepsis treatment. We have demonstrated that a combination of NT$^{(+)}$ and a moderate IMI antibiotic treatment yielded a 100% survival in CLP mice in repeated studies in CLP septic mice with different sex and ages. The analysis for both cytokines and inflammatory signaling pathways revealed the significantly reduced local, systemic, and remote organ inflammation, which correlated well with the reduced organ failure, and strongly supports the survival benefit by NT$^{(+)}$/antibiotics treatment. TD NT resins are readily incorporated into the standard clinical care for sepsis treatment, e.g., local immune intervention or hemoperfusion therapy. If tested effective, it can also be applied to treat other critical illness with high risk of cytokine storm, for example, trauma, burn, and severe cardiac surgery.

## Methods

**Materials**. All chemicals were used as received, unless otherwise specified. Rink Amide-MBHA resin (HCRAm 04-1-1) was ordered from Nankai HECHENG S&T Co., Ltd (Tianjin, China). Amino PEGA resin (Novabiochem®, Darmstadt, Germany) was obtained from EMD Millipore (Billerica, MA). (Fmoc)-Lys(Boc)-OH, (Fmoc)-Lys(Fmoc)-OH, and trifluoroacetic acid (TFA) were obtained from Chem-Impex International, Inc. (Wood Dale, IL). (Fmoc)-Arg(Pbf)-OH was purchased from AnaSpec Inc. (San Jose, CA). N,N'-diisopropylcarbodiimide (DIC), N-hydroxybenzotriazole (HOBt), succinic anhydride, 4-dimethylaminopyridine (DMAP), and N,N-dimethylformamide, anhydrous (DMF, 99.8%) were received from Acros Organics (Belgium, NJ). Polymyxin B Sulfate, Polymyxin B-Agarose (P1411), LPS from *Escherichia coli* (L4130), and *Pseudomonas aeruginosa* (L9143) were purchased from Sigma-Aldrich (St. Louis, MO). Polylysine–cellulose resin (Pierce™) was purchased from Thermo Scientific (Rockford, IL). Limulus amebocyte lysate (LAL) endotoxin quantification kit was purchased from Pierce™ (Thermo Scientific™, IL) and performed following the manufacturer's instructions. ELISA kits were purchased from companies for direct use (e.g., HMGB-1: Cat. #: NBP2-62767 from Novus Biologicals, IL-1β: Cat. # BMS6002 from Invitrogen, IL-6 Cat. #: BMS603-2 from Invitrogen, and TNF-α: Cat. #: BMS607HS from Invitrogen); ECL (Cat. #: 34580) was purchased from Thermo Fisher Scientific. HRP-conjugated secondary antibody (Cat. #: sc-516102, 1:4,000) was purchased from Santa Cruz biotechnology, Santa Cruz, CA. PVDF membrane (Cat. #: IPVH00010) from Millipore Co., Ltd. and Bio-Rad protein assay (Cat. #: 50000001) from Bio-Rad Laboratories were used. Antibodies for NF-κB (Cat. #: sc-8008, 1:200), P-iκB-α (Cat. #: sc-8404, 1:200), and β-actin (Cat. #: sc-47778, 1:500) were purchased from Santa Cruz Biotechnology (Santa Cruz, CA).

**Instrumental methods**. Matrix-assisted laser desorption/ionization time-of-flight mass spectrometry (MALDI-TOF MS) spectra were collected on a Bruker Autoflex

III system equipped with a Smart beam II laser source and acquired in positive, reflector mode. $^1$H NMR spectra were recorded on a 600-MHz Bruker AVANCE NMR spectrometer. Transmission electron microscopy (TEM) characterization of nanoparticles was performed on JEOL JEM-1400 operated at 80 kV. Samples were prepared on glow-discharged carbon-coated copper grids (CF300-CU, 300 mesh, Electron Microscopy Sciences). The hydrodynamic sizes of nanoparticles were acquired by dynamic light-scattering (DLS) measurement using a particle analyzer (Microtrac Zetatract). Confocal microscope (Nikon) images were acquired in z-stack mode having sequential optical x–y sections taken with a z interval at 5 μm.

**Solution-phase telodendrimer synthesis.** Telodendrimers bearing both guanidine and hydrophobic groups were initiated from methoxy-terminated amino PEG, MeO–PEG–NH$_2$ (Mw: 5 kDa) following a published procedure[25]. N-terminal-protected lysine was used to synthesize the branched scaffold of polylysine dendrons using HOBt/DIC as coupling reagents in anhydrous DMF at room temperature. All reagents are in 3 equiv. stoichiometric excess relative to the primary amine in the intermediates tethered on PEG. The completion of reactions was monitored by the chromogenic ninhydrin tests to probe the consumption of primary amine: dark blue indicates the presence of primary amine; yellow color indicates the completion of amine coupling. Fmoc-protecting group was removed by the treatment of 20% 4-methylpiperidine in DMF for 30 min. Pbf-protecting group was deprotected in the presence of TFA/DCM (50/50, v/v) for 2 h. PEGylated intermediates and telodendrimer products were precipitated from the solution after the reaction by the addition of tenfold volume of ice-chilled ether and collected by centrifugation at 1950×g. Precipitates were rinsed by ice-chilled ether three times and dried in vacuum for further reaction. The final telodendrimer was purified by dialysis and lyophilized for further characterization and applications.

**Solid-phase synthesis of LPS-binding moieties.** Starting from TentaGel (TG), or PEGA, or PVA–PEG resin, (Fmoc)-Lys(Fmoc)-OH, and Fmoc-Oligo (ethylene glycol)-COOH linker were coupled sequentially following the standard peptide synthesis procedures (Supplementary Fig. 6). DIC and HOBt were used as catalytic coupling reagents. All reactants were in threefold excess with respect to the amine functional group on resin. After second-generation dendritic oligolysine synthesis, (Fmoc)-Arg(Pbf)-OH was used to introduce the third layer of oligolysine on acid-labile Rink resin and acid-inert resin (TG, PEGA, and PVA–PEG) to introduce the orthogonally protected amine groups for charge and hydrophobic moiety conjugation. De-Fmoc was carried out in 20% 4-methylpiperidine DMF solution for 30 min. Pbf- protecting group was removed in the presence of TFA/DCM (50/50, v/v) for 2 h. After the completion of each step reaction, residual reactants were removed under vacuum and washed with copious solvents of DMF, DCM, and MeOH sequentially. The LPS-binding hydrophobic building blocks were conjugated on the α-amine on the arginine after de-Fmoc step following the insertion of a triethyleneglycol linker molecule via the standard peptide synthesis procedure. LPS-binding dendron synthesized on Rink resin will be treated with TFA/TIS/H$_2$O (95/2.5/2.5, v/v/v) cocktail to cleave the arginine-protecting group to release guanidine group, and simultaneously cleave the whole dendron from Rink resin into solution. In parallel, TG (PEGA or PVA–PEG) resin modified by LPS-binding dendron was washed intensively and readily available for LPS adsorption and removal.

**Electrophoresis assays.** The binding capacities of the telodendrimers with LPS and/or bovine serum albumin (BSA) were studied and compared with PMB binding using electrophoresis assay. The electrophoresis was carried out in 1.5% agarose gel (Tris-borate-EDTA (TBE) buffer) at constant current of 20 mA for 2 h. The gel was imaged by a Bio-Rad Universal Hood II Imager (Bio-Rad Laboratories, Inc.) under SYBR Green modes or photographed under UV illumination.

**Fluorescent polarization assays.** The fluorescence polarization (FP) was measured on the Multi-Mode Microplate Reader (Synergy$^{TM}$ 2, Biotek, VT) equipped with dichroic mirror (510 nm) and polarizing filter. The measurements were carried out on black flat-bottom 96-well plates (Nunclon$^{TM}$ Surface, Roskilde, Denmark). The FP of LPS-FITC was recorded at excitation and emission filter of 485/20 nm and 528/20 nm, respectively. The experiments were performed in triplicate.

**Resin-binding assay for biological molecules.** The desired amount of resin was weighed, hydrated, and incubated with fluorophore-labeled biological molecules, including LPS, BSA, myoglobin (Mb), lysozyme, α-lactalbumin, and TNF-α in medium (PBS or FBS or whole blood) for the defined time period. Then, the incubation solutions were collected for the fluorescence measurement by microplate reader (BioTek Synergy 2), and the resins were washed with PBS three times. BSA-RB and/or LPS-FITC-bound resins were then visualized under fluorescence microscopes.

**Protein elution and MALDI-TOF MS analysis.** Protein-adsorbed nanotrap resins were washed with PBS intensively and drained to dryness using a centrifuge tube with filter (0.22 μm, Corning$^{TM}$ Costar$^{TM}$ Spin-X$^{TM}$). The resin was then incubated with two times volume of 6 M guanidine or 8 M urea solution at room temperature overnight. The elution was collected and spotted on MTP 384 target plate (Bruker Daltonics) after mixing with CHCA matrix in 50% acetonitrile with 0.1% TFA. The spectra were collected using a linear mode.

**LPS attenuation.** Macrophage RAW 264.7 cells (ATCC) were cultured in complete DMEM medium supplemented with 10% FBS, 100 U/ml penicillin, and 100 U/ml streptomycin at 37 °C using a humidified 5% CO$_2$ incubator. Cells were plated in 96-well plates at a density of $2 \times 10^4$ cells/well. Stock solution of LPS (10 μg per mL) derived from *Pseudomonas aeruginosa* (L9143, Sigma-Aldrich) was pretreated with nanotrap beads for overnight incubation before being added into macrophage cell culture. The untreated stock LPS solution was directly added to the cell culture to a final LPS concentration of 500 ng per mL as a control for cytokine production comparison. After 24-h incubation, cell medium was collected, and the supernatant was obtained by centrifugation for TNF-α production analysis using mouse TNF-α ELISA Ready-SET-GO assay (eBioscience$^{TM}$).

**Sepsis model induced by CLP in mice.** Specific-pathogen-free BALB/c (8–10 weeks or 11 months, both sexes) purchased from Charles River (USA) were kept under pathogen-free conditions (12-h light–dark cycle, 22 ± 2 °C and 60% air humidity) according to the AAALAC (Association for Assessment and Accreditation of Laboratory Animal Care) guidelines, and were allowed to acclimatize for at least 4 days before any experiments. All animal experiments were performed in compliance with the institutional guidelines and according to the protocol approved by the Committee for the Humane Use of Animals of State University of New York Upstate Medical University.

For CLP procedure, mice were anesthetized using intraperitoneal ketamine/xylazine (100 mg/kg ketamine, 10 mg/kg xylazine) injection. After adequate anesthesia, the lower quadrants of the abdomen were shaved and the surgical area was disinfected. A longitudinal midline incision was made using scissors to extend the incision into the peritoneal cavity. After fascial, intramuscular, and peritoneal incision, the cecum was located and exteriorized. The cecum was tightly ligated with a 1.0 suture (COATS, ART 230 A) at about 1.3 cm to the distal end, and was perforated into two holes with a 22-gauge needle. One hole is 0.5 cm to the distal end of the cecum and the other hole is 0.5 cm near the ligation side. The cecum was then gently squeezed to extrude a small amount of feces (about 1 mm³) from the perforation sites. The cecum was returned to the peritoneal cavity, and the peritoneum and skin are closed with 5.0 silk sutures. In sham group, mice were operated following the same protocol without CLP procedure. After operation, mice were resuscitated with 1 mL of warmed saline immediately. The animals were returned immediately to a cage with exposure to an infrared heating lamp for 30 min–1 h, until recovery from anesthesia. Mice were provided with free access to food and water in the bottom of the cage. Buprenorphine (0.05 mg per kg, SQ) was injected for postoperative analgesia every 12 h.

**Sepsis treatments in CLP mice.** Following the above standard CLP procedure, resuscitation, and pain management procedure post operation, CLP mice were randomly assigned in groups (n = 8–10) for the treatments with NT resins, antibiotics, and saline according to the specific experimental design: (1) initial treatment studies in Fig. 5d: 150 μL of blank PEGA resin or NT$^{(+)}$ PEGA(Arg$_4$C17)$_4$ in 1× PBS or saline control was injected into the peritoneal cavity of CLP mice (n = 10, female BALB/c, 8–10 weeks) through surgery wound right after CLP procedure before wound closure. Animals were monitored frequently for 3 days for mortality comparison. (2) Time and charge effects of NT resins in Fig. 5e: 150 μL of NT$^{(+)}$ PEGA(Arg$_4$C17)$_4$ or NT$^{(−)}$ PEGA(OAC17)$_4$ resins in 1× PBS were injected into the peritoneal cavity of CLP mice (n = 8–10, female BALB/c, 8–10 weeks) using blunt needle through the surgery wound sites at 0 h, 3 h, or 8 h, respectively, after CLP procedure under Fluriso$^{TM}$ inhalation. Then the incision was re-closed with 3 M Vetbond. Mice were monitored frequently for survival studies over 6 weeks, and body weight was monitored over time. About 100 μL of blood was collected from the tail vein at different timepoints for blood cell counting, and body temperature were taken by a digital infrared thermometer at the abdomen at different times (5 h, 11 h post surgery, daily for the first week, and once per week afterward). (3) Combination therapy in Fig. 6b: 3 h after CLP procedure, mice (n = 8–10, female BALB/c, 8–10 weeks) were divided randomly for i.p. injections with 150 μL of saline or 150 μL of NT$^{(+)}$ PEGA(Arg$_4$C17)$_4$ or NT$^{(−)}$ PEGA(OAC17)$_4$ resins with/without i.p. injection of antibiotics imipenem/cilastin (50/50 mg per kg body weight) every 8 h up to 3 days. Mice were monitored frequently for survival studies, and body weight was monitored over time. About 100 μL of blood was collected from the tail vein at different timepoints for blood cell counting, and body temperature was taken by a digital infrared thermometer at the abdomen daily for the first week, and once per week afterward. (4) Following the same procedure, male CLP mice (n = 8–10, female BALB/c, 8–10 weeks) were treated with saline, IMI, NT$^{(+)}$, and NT$^{(+)}$/IMI for survival observations for 6 weeks. Body weight, body temperature, and blood counts were monitored over time (Supplementary Fig. S14). (5) Similarly, aged female BALB/c mice (n = 5, 11 months) were used to create CLP sepsis models, and were treated with saline, antibiotics, IMI, or NT$^{(+)}$/IMI for survival observations for a week (Supplementary Fig. 15).

**Cytokine analysis and histological examination**. Septic mice were induced by the same CLP procedure ($n = 5$, female BALB/c, 8–10 weeks) and were treated with the identical reagents as the combination therapy in Fig. 6i, e. saline, antibiotic IMI, NT$^{(+)}$, NT$^{(+)}$/IMI, NT$^{(-)}$, and NT$^{(-)}$/IMI, respectively. In addition, a sham group was included with a laparotomy procedure with cecum taken out and put back into the abdomen before wound closure. Mice were sacrificed 24 h post CLP under anesthesia (ketamine: 100 mg per kg, xylazine: 10 mg per kg, IM). The peritoneal cavity was rinsed with 600 μL of PBS to collect peritoneal lavage for cytokine analysis. Blood was collected from the inferior vena cava. Peritoneal lavage and plasma were stored at −80 °C for resin treatment or cytokine analysis. Cytokines from plasma and peritoneal lavage were measured by Elisa or multiplex immune assay (LEGENDplex™ Mouse Inflammation Panel (13-plex), 740150) by flow cytometry according to the manufacturer's instruction. Heart, liver, lung, kidney, spleen, and intestine were harvested. Part of tissue was processed for protein extraction for NF-κB and HMGB-1 analysis via western blot. The rest of tissue was fixed by 10% neutral buffered formalin or frozen in OCT cryo-embedding medium for histologic study. In order to analyze pulmonary structure–function relations, right-side lung lobe was fixed by formalin infusion into the cannulated main bronchus and was immersed in a container of formalin for at least 24 h. Then, the fixed lung tissue was embedded in paraffin for sectioning (5 μm) and then stained with hematoxylin and eosin (H&E) for histopathology analysis.

**Characterization of inflammatory activation**. Small intestine and liver were collected after mice were sacrificed. The gut was flushed with ice-cold PBS. The mucosa of the small bowel segments was scraped using a microscope slide. Mucosal scrapings and liver were homogenized using tissue grinder in RIPA lysis and extraction buffer. Bio-Rad protein assay was used to determine protein concentration (Bio-Rad Laboratories, 50000001). In total, 0.1 mg in 100 μl of protein lysate or 50 μl of serum (1:100 dilution) was used for ELISA assay. Biomarkers were measured using commercial ELISA kits according to the manufacturer's instructions (HMGB-1: Novus Biologicals, NBP2-62767; IL-1β: Invitrogen, BMS6002; IL-6: Invitrogen, BMS603-2; TNF-α: Invitrogen, BMS607HS). About 20 μg of proteins were loaded and separated by SDS-PAGE, and then transferred to PVDF membrane (Millipore, Cat #: IPVH00010, lot #: R9KA75004). The membranes were blocked with 5% nonfat milk in Tris-buffered saline plus 0.5% Tween-20 for 1 h at room temperature and incubated overnight at 4 °C with the indicated primary antibodies, respectively, for NF-κB (Cat #: sc-8008, lot #: F2019, mouse monoclonal IgG$_1$, 1:200, Santa Cruz, sc-8008), P-iκB-α (Cat #; sc-8404, mouse monoclonal IgG$_{2b}$, Lot #: 0718, 1:200, Santa Cruz, sc-8404), and β-actin (Cat #: sc-47778, lot #: B0719, mouse monoclonal IgG$_1$, 500, Santa Cruz). After primary antibody incubation, the blot was incubated with HRP-conjugated secondary antibody (Cat #: sc-516102, lot #: J0919, 1:4000, mouse IgG kappa binding protein, Santa Cruz, sc-516102) for 1 h at room temperature. Antibody–antigen complexes were visualized with ECL (Thermo Fisher Scientific, 34580) and analyzed quantitatively by densitometry with Image J software (1.52a). The relative density of immune-reactive bands was normalized to the density of the corresponding β-actin bands.

**Statistical analysis**. In vitro experiments were conducted in triplicate, and data are presented as mean ± SEM. Nonparametric, one-tailed Student's t test was performed for comparison of two groups. One-way and two-way analyses of variance (ANOVAs) with Newman–Keuls post hoc correction (GraphPad Prism 8) were used for multiple-group analyses. The log-rank (Mantel–Cox) test (GraphPad Prism 8) was used to compare the difference in Kaplan–Meier survival plot between different groups. The level of statistical significance was set with $P < 0.05$.

**Reporting summary**. Further information on research design is available in the Nature Research Reporting Summary linked to this article.

## Data availability

All data associated with this study are present in the paper or the Supplementary Materials. Additional relevant information is available from the authors upon reasonable request. Source data are provided with this paper.

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

## Acknowledgements

We thank Professor Richard Wojcikiewcz and Prof Gary Nieman for helpful discussion and support. We thank Professor Kit S. Lam for suggestions in evaluation of protein penetration in hydrogel resin via protease assay. We thank Dr. Osama Abdel-Razck for help in animal surgery. We thank Professor Shengle Zhang for help in pathology studies. We greatly acknowledge the financial support from NIH/NIGMS 1R01GM130941-01, NIH/NHLBI 1R01HL139824-01, New York Fund for Innovation in Research and Scientific Talent (FIRST), and Maureen T. O'Hara TEAL THERE's A CURE and Christine Schoeck Blakely Ovarian Cancer Research Foundation.

## Author contributions

C.S., X.W., and L.W. conducted the experiments and analyzed the data. Q.M., G.D., N.C., and M.D. partially performed experiments and/or analyzed the data. G.W. and C.R. assisted in experimental design and paper editing. J.L. designed experiments, interpreted the data, and prepared the paper.

## Competing interests

C.S., L.W., and J.L. are inventors on a related patent submitted by State University of New York Upstate Medical University (International Application No. PCT/US2018/0497, published March 14, 2019). The remaining authors declare no competing interests.
