## [Peer Review File · Nature Communications]

Reviewers' Comments:

Reviewer #1:

Remarks to the Author:

Wang et al reported a comprehensive study on a novel nanotrap resins that leverage different properties including charges, sizes, and hydrophobicity to remove LPS and cytokines, as a way to treat sepsis. If successful, it could make a significant impact in managing sepsis that remains to be a major unmet clinical need. However, my initial excitement was significantly dampened as the results demonstrated the new system only exhibited moderate improvement compared to commercial controls, as I outlined below along with other comments. In light of this, the impact of this work is likely minimal, unless the utility of this new system to improve sepsis outcome in a clinically relevant animal model can be demonstrated, which is lacking.

- 1) In Fig. 3D, when evaluating removing LPS in clinical concentrations using the new resins compared to commercial controls, the improvement is minimal, i.e. 95% vs 90%. Authors may therefore need to demonstrate this small improve can translate to clinically relevant outcomes, otherwise, this work may not possess the impact as claimed.
- 2) Along this line, in Fig 7B, control commercial resins should be performed and shown head-to-head.
- 3) To further assess the clinical impact, in Fig 6, protein capture, take 2 hours, not sure if this time frame is clinically relevant in a hemodialysis system. do not we need faster clearance?
- 4) Fig 2 has significant issues: a) it says "TG-(ArgVE)₄ resin could adsorb FITC-LPS efficiently after a short incubation (2-3 min) and beads were light up intensively (Fig. 2B)". however, Images did not show that, in fact, they appeared all dark. b) "However, BSA with larger size can only be captured on the surface of both TG and PVA-PEG resins." Images did not show BSA can be captured on the surface of TG resins, which is a surprise. c) It is unclear why the resins exhibited dramatically different sizes in Fig C and D. Are their sizes intrinsically heterogeneous? d) Fig 2E, symbol μm did not show properly. e) The way images are presented in some of the images in Fig 2E and Fig S8, i.e. only showed small fractions of resin rather than the whole make it difficult to assess molecule diffusion.
- 5) It is unclear how the "removal efficiency" is determined in general in the paper? It seems it was determined based on fluorescence reduction from the mixture of target molecules and complex media such as blood. Given the autofluorescence from the matrix, the authors need to clearly describe how the "removal efficiency" is calculated, e.g. was any normalization done?
- 6) In several figures NH₂ in chemical structures including those in SI, 2 should be in the subscript form

Additional minor issues:

- 7) Fig 1B, why negative LPS-FITC showed on two different bands?
- 8) on a related note, in Fig 3C, why LPS derived from E coli and P. aeruginosa behave so differently in resin capture? are their molecular or structural differences between these two types of LPS?

Reviewer #2:

Remarks to the Author:

In the present study, Wang et al developed a novel "octopus-like" flexible telodendrimer (TD) nanotrap, and tested its efficacy of capturing a broad range of biomacromolecules, including bacterial endotoxins and small proinflammatory and anti-inflammatory cytokines from macrophage cell cultures or plasma of septic animals. They reported that this nanotrap could capture macromolecules via multivalent electrostatic and hydrophobic interactions, and could be conjugated to size-exclusive hydrogel column in order to selectively remove macromolecules with

certain molecular weight cutoffs. The authors provided sufficient evidence to support its improved affinity to bacterial endotoxins (as opposed to the classical polymyxin B resin), as well as its feasibility to remove several small cytokines from biological samples. Compared with polymyxin B- and antibody-based unimodal hemo-sorbents, this modified nanotrap-based hemoperfusion might provide more advantages for possible application in future clinical management of human sepsis. However, its therapeutic potential has not yet been tested in any animal models of inflammation, which significantly reduces the significance of this seemingly preliminary study.

1. It is critically important to assess the efficacy of this nanotrap-based hemoperfusion in an animal model of lethal sepsis or other inflammatory diseases.
2. Many small proinflammatory cytokines could either bind to other proteins to form large complex, or be enclosed in microvesicles in the plasma of septic animals or patients. Thus, size-exclusion might not be feasible to remove the cytokines captured by other plasma proteins or enclosed by plasma microvesicles.
3. The design of some experiments [e.g., the prolonged incubation of resin with LPS-containing blood for an extended period of time (2 h), as indicated in Fig. S9B) was not clinically relevant, as the fast blood flow (e.g., 100-200 ml/min) of hemoperfusion will not accommodate the aforementioned time-consuming capturing process.
4. In the abstract, it is not clear whether the authors were referring the "gene molecules" as proteins or DNAs?
5. In addition, LPS is also considered as a pathogen-associated molecule pattern (PAMP) molecule.

Reviewer #3:

Remarks to the Author:

This manuscript reports a telodendrimer based adsorption mat for sepsis biomacromolecular markers. Overall, I feel that it is an interesting study which in my opinion should be published after revisions as a full paper in a journal such as Biomaterials, and not in Nature Communications.

The novelty in this study is not in the design of the hybrid architecture, as it has been reported and studied in detail earlier ("novel flexible telodendrimer nanoplatform" already claimed to be a "novel finding" in their earlier papers, and its ability to encapsulate proteins demonstrated), but may be in extending this work to include spread of tuned charges that facilitate adsorption via amphiphilic nature of the designer substrate. Although the overall design may be considered to be borrowed from Polymyxin B, the authors do not address the need for this complex architecture which may limit translation into a clinical setting, compared to other simpler and easily assembled polymeric architectures with similar multivalency in charge spreading. Mentioned sporadically but nowhere explained is the claim of octopus structure of the trap mats (except for maximizing conformational entropy??), and the necessity to have a hydrophilic tail in the design mode of telodendrimers. Is the self-assembly of such architectures with a CMC of 1-2 micro molar of significance, as it is ultimately forming a complex with LPS for adsorption purposes? Are the nanoassembled structures first formed which subsequently open and reassemble upon interaction with LPS? For maximum binding in TD-LPS nanocomplex, and its subsequent stability clearly suggests that the dendritic arms need to be extended (also evident from the data provided), so what is the advantage of the telodendrimer architecture and its self-assembly? What is the immobilization efficiency of PMB on the resin and how does it compare with the telodendrimer nanotrap? This will play a role and help address the issue of superiority of the telodendrimer nanotrap in LPS removal. Diffusion characteristics in PMB and telodendrimer immobilized resins are expected to be different on the overall structure of the adsorbed species. LPS removal is more efficient with the telodendrimer model but cytokine removal efficiencies were only somewhat improved from PMB based resin. This is again related to overall structural differences between the two.

This is the reason I repeat, that the novelty of the design itself is lacking in the current setting, and established more in their earlier similar explorations using this architecture. This question is of

more merit when the efficiency of these structures is claimed to be through immobilization and selective adsorption on "size-exclusive hydrogel resins". There is no detailed evaluation of the "rational selection of amphiphilicity" and its influence in tuning the efficacy. The study makes a claim of "fine-tuned all in one hemoperfusion", but without listing structure-property relationships of this. The criticism for commercially available Toraymyxin and Cytosorb is misplaced as the claims made here have not been tested clinically and may lead to similar unfortunate outcomes.

Reviewers' comments:

Reviewer #1 (Remarks to the Author):

Wang et al reported a comprehensive study on a novel nanotrap resins that leverage different properties including charges, sizes, and hydrophobicity to remove LPS and cytokines, as a way to treat sepsis. If successful, it could make a significant impact in managing sepsis that remains to be a major unmet clinical need. However, my initial excitement was significantly dampened as the results demonstrated the new system only exhibited moderate improvement compared to commercial controls, as I outlined below along with other comments. In light of this, the impact of this work is likely minimal, unless the utility of this new system to improve sepsis outcome in a clinically relevant animal model can be demonstrated, which is lacking.

Response: We really appreciate your recognition on the significance and potential clinical impact of our technology. It was a reasonable request to include *in vivo* efficacy. After last submission, we have conducted comprehensive *in vivo* studies in mouse sepsis models induced by cecum ligation and puncture (CLP). We have demonstrated the significant survival benefit through attenuating hyperinflammation via our nanotrap treatment. Further, 100% survival in CLP sepsis were observed in the repeated studies in mice with different genders and ages by the combination of our nanotrap treatment with moderate antibiotic treatment.

1) In Fig. 3D, when evaluating removing LPS in clinical concentrations using the new resins compared to commercial controls, the improvement is minimal, i.e. 95% vs 90%. Authors may therefore need to demonstrate this small improve can translate to clinically relevant outcomes, otherwise, this work may not possess the impact as claimed.

Response: It is not surprising that commercial resin with positive charges can also adsorb significant LPS at very low ng/mL concentrations with the excessive amount of resin via surface charge interactions. Although LPS clearance by our TD nanotrap resin were not dramatically better than commercial resin at low LPS concentration, i.e. 95% vs 90%, which can't directly reflect the potential *in vivo* efficacy. Instead, high capacity and fast adsorption are required for effective LPS removal from large volume blood during hemoperfusion. As shown in **Figure 3**, our TD nanotrap resin showed much higher efficiency and capacity for LPS adsorption at high LPS concentration. In addition, the attenuation of multiple inflammatory mediators concurrently, not only LPS, is critical to prevent hyperinflammation and organ failure in sepsis as shown in the *in vivo* studies. Therefore, LPS clearance at low concentration was removed from the resubmission to avoid the misleading.

2) Along this line, in Fig 7B, control commercial resins should be performed and shown head-to-head.

Response: The commercial resins, e.g. Argarose-PMB and Cellulose-P(ϵ -lys) are designed for *in vitro* LPS adsorption and not for protein adsorption and not hemocompatible with significant cell adhesion. Therefore, it was not compared for cytokine adsorption.

3) To further assess the clinical impact, in Fig 6, protein capture, take 2 hours, not sure if this time frame is clinically relevant in a hemodialysis system. do not we need faster clearance?

Response: It is understood that fast clearance is preferred for hemoperfusion therapy. As shown in **Fig 4B**, protein can be efficiently adsorbed via incubation within 30 min and even for a few minutes under agitation. The clinical hemoperfusion therapy is generally set for 4 hours' continuous treatment and a repeat treatment can be followed. Therefore, protein capture assay was designed for 2 h incubation to allow different resins to function or equilibrium, which is also within the clinical window of hemoperfusion treatment.

4) Fig 2 has significant issues: a) it says "TG-(ArgVE)4 resin could adsorb FITC-LPS efficiently after a short incubation (2-3 min) and beads were light up intensively (Fig. 2B)". however, Images did not show that, in fact, they appeared all dark. b) "However, BSA with larger size can only be captured on the surface of both TG and PVA-PEG resins." Images did not show BSA can be captured on the surface of TG resins, which is a surprise. c) It is unclear why the resins exhibited dramatically different sizes in Fig C and D. Are their sizes intrinsically heterogeneous? d) Fig 2E, symbol μm did not show properly. e) The way images are presented in some of the images in Fig 2E and Fig S8, i.e. only showed small fractions of resin rather than the whole make it difficult to assess molecule diffusion.

Response: I am sorry that there must be some issues of displaying during your review. From the manuscript submitted online, there was no problem for the images as shown below: (a) as shown in the following figure 2B, TG(ARGVE)4 resin was indeed much more brighter than TG-PMB resin for both with/without free PMB; (b) As shown in Figure 2C, FITC-BSA was indeed only adsorbed on the surface of TG(ArgVE)4 resin; (C) These hydrogel resins were purchased or synthesized in lab with different materials, therefore, we didn't control the size distribution of the resin; (D) Bead pictures shown in Figure 2D was to illustrate the protein diffusion process, therefore no scale bar was inserted; (E) The diffusion distance were analyzed with software on the digital pictures for numbers of beads, the pictures showed in Figure 2E and Figure S8 were intended to illustrate the phenomena of gradual bead light up.

Some pictures for protein uptake in different resins were not shown in the new version of manuscript due to the limited space. We skip the studies on resin optimization and comparison, and directly reported the results on the optimal PEGA resin to make the manuscript more focus and easy to follow.

5) It is unclear how the "removal efficiency" is determined in general in the paper? It seems it was determined based on fluorescence reduction from the mixture of target molecules and complex media such as blood. Given the autofluorescence from the matrix, the authors need to clearly describe how the "removal efficiency" is calculated, e.g. was any normalization done?

Response: The removal efficiency was measured by the fluorescent reduction of solutions of FITC labeled LPS or proteins before and after bead treatment. Since the background fluorescent of serum protein is generally very low compared with fluorescent molecule FITC, therefore, the normalization was not conducted. Also MALDI-ToF MS spectrum was also applied to qualify model protein removal efficiency. For cytokine adsorption, ELISA and multiplex immune assay were used for quantification.

6) In several figures NH₂ in chemical structures including those in SI, 2 should be in the subscript form

Response: Thank you for pointing it out. We have corrected as we can find out.

Additional minor issues:

7) Fig 1B, why negative LPS-FITC showed on two different bands?

Response: As LPS from different bacteria strains may have different chemical structures, molecular weight and charge density, therefore migrates differently in electrophoresis

8) on a related note, in Fig 3C, why LPS derived from E coli and P. aeruginosa behave so differently in resin capture? are their molecular or structural differences between these two types of LPS?

Response: Similar to the question above, LPS from different bacteria strains may have different structure, molecular weight and charge density, which may affect the diffusion and binding affinity with nanotrap resin. However, we would say that the adsorption efficiency for both LPS in nanotrap resins were similarly effective around 98-99%.

Reviewer #2 (Remarks to the Author):

In the present study, Wang et al developed a novel “octopus-like” flexible telodendrimer (TD) nanotrap, and tested its efficacy of capturing a broad range of biomacromolecules, including bacterial endotoxins and small proinflammatory and anti-inflammatory cytokines from macrophage cell cultures or plasma of septic animals. They reported that this nanotrap could capture macromolecules via multivalent electrostatic and hydrophobic interactions, and could be conjugated to size-exclusive hydrogel column in order to selectively remove macromolecules with certain molecular weight cutoffs. The authors provided sufficient evidence to support its improved affinity to bacterial endotoxins (as opposed to the classical polymyxin B resin), as well as its feasibility to remove several small cytokines from biological samples. Compared with polymyxin B- and antibody-based unimodal hemo-sorbents, this modified nanotrap-based hemoperfusion might provide more advantages for possible application in future clinical management of human sepsis. However, its therapeutic potential has not yet been tested in any animal models of inflammation, which significantly reduces the significance of this seemingly preliminary study.

Response: Really appreciate your positive evaluation on our technology. To address your concerns for the in vivo efficacy, we have accomplished comprehensive treatment studies as reported in the new manuscript.

1. It is critically important to assess the efficacy of this nanotrap-based hemoperfusion in an animal model of lethal sepsis or other inflammatory diseases.

Response: In order to understand the molecular consequence and efficacy of the nanotrap resin treatment in immune modulation, we applied lethal CLP sepsis mouse model and treated animal *in situ* with bead adsorption. We have demonstrated the significant survival benefit through attenuating hyperinflammation via our nanotrap treatment. Further, 100% survival in CLP sepsis were observed in the repeated studies in mice with different genders and ages by the combination of our nanotrap treatment with moderate antibiotic treatment.

2. Many small proinflammatory cytokines could either bind to other proteins to form large complex, or be enclosed in microvesicles in the plasma of septic animals or patients. Thus, size-exclusion might not be feasible to remove the cytokines captured by other plasma proteins or enclosed by plasma microvesicles.

Response: Thank you for your input and it is reasonable concern that the oligomerization, association and sequestration of inflammatory molecules may prevent the effective scavenging by nanotrap resins due to the size-exclusive effects. On the other hand, it is not intended to remove all inflammatory mediators, rather than attenuate the excessive inflammations, e.g. hyperinflammation to prevent multiple organ failure. It is believed that the complete inhibition of inflammation is detrimental for sepsis treatment,¹⁻³ which has also shown in our optimization studies **Fig. 5E** in the new manuscript.

It is reported that cytokines can also be associated with or encapsulated in extracellular vesicle (EV) to increase the local concentration upon release proximal to the target cells for basal level immune regulation. It also was reported that the EV-associated cytokines are downregulated upon immune stimulation; Instead, the production of the free form cytokines are significantly upregulated during inflammation.⁴ The associated inflammatory molecules, e.g. LPS and cytokines are inactive and need to be released from association to stimulate/mediate inflammation, thereafter, subject to NT adsorption. For instance, TNF- α can be significantly removed by NT resins in vitro (As shown in **Fig. 4 G&H** in previous submission and **Fig. 5B** in new submission) in spite of trimer formation of TNF- α .

3. The design of some experiments [e.g., the prolonged incubation of resin with LPS-containing blood for an extended period of time (2 h), as indicated in Fig. S9B) was not clinically relevant, as the fast blood flow (e.g., 100-200 ml/min) of hemoperfusion will not accommodate the aforementioned time-consuming capturing process.

Response: Fast clearance of protein with the optimal resin has been shown in Fig 4B with >90% of protein adsorbed within 30 min, even after a few minutes' incubation. It is believed that the efficiency of adsorption by passing through the column packed with resins will be more effective than bead incubation with shaking. The clinical hemoperfusion therapy is generally set up for 4 hours continuous treatment with the repeated treatment can be followed. Therefore, protein capture assay was designed for 2 h incubation, which is within the clinical settings.

4. In the abstract, it is not clear whether the authors were referring the "gene molecules" as proteins or DNAs?

Response: The genetic DAMP molecules refer to cell-free DNA and RNA fragments, which stimulate immune reactions through TLR binding.

5. In addition, LPS is also considered as a pathogen-associated molecule pattern (PAMP) molecule.

Response: Yes, LPS is one of the most important PAMP molecules, which is clarified in the manuscript. Because it is so potent in triggering innate immune reactions, therefore, were separately emphasized.

Reviewer #3 (Remarks to the Author):

This manuscript reports a telodendrimer based adsorption mat for sepsis biomacromolecular markers. Overall, I feel that it is an interesting study which in my opinion should be published after revisions as a full paper in a journal such as Biomaterials, and not in Nature Communications.

Response: We appreciate the thoughtful questions which help us to further improve the development of this technology. In summary, the reviewer focused on the following key issues (1) innovation and novelty; (2) Comparison of the telodendrimer design over conventional polymers or PMB with charge and hydrophobic components. Here we first address these overarching questions, then we will answer your technical questions in detail.

(1) Novelty and innovation: We agree the novelty of this study is not the chemical design of the TD, which was reported in our previous study for protein delivery. Here, we demonstrated the application of this TD nanoplatform in attenuating endotoxin and other DAMPs and PAMPs biomolecules in addition to cytokine protein molecules. The most important innovation is the first time to develop a nanoplatform to target multiple inflammatory mediators, which is an unmet clinical need and essential to attenuate hyperinflammation in sepsis to reduce mortality. The conceptual innovations include: (a) enhanced LPS adsorption with superior efficiency to the gold standard LPS-binding PMB; (b) the combination of versatile nanotrap with the size exclusive effects in hydrogel allows for the adsorption of broad spectrum of inflammatory mediators; (c) the first time to discover the charge disparity of cytokines, i.e. majority proinflammatory cytokines are negatively charged and most anti-inflammatory cytokines are positively charged. It allows for effective and precise immune modulation using TD nanotrap approach by decorating specific charge moieties to adsorb proinflammatory cytokines to attenuate hyperinflammation or adsorb anti-inflammatory cytokines to reverse immune suppression based on the immune status of patients; In the new version of manuscript, we clearly demonstrated the importance of the precise immune modulation via TD nanotrap with different charges and different intervention schedule in treating CLP mouse sepsis models. (d) In the new submission, we have also demonstrated dramatic survival benefits by TD nanotrap intervention in CLP sepsis, and 100% survival were observed in repeated treatments in CLP septic mice in combination of antibiotics. This study verifies the importance of the attenuation of excessive inflammation in sepsis treatment, which however is unmet in the clinic.

(2) The rationale and evidence of dendritic TD design over PMB and conventional linear polymers possessing charge and hydrophobic moieties:

Firstly, we would like to compare TD with PMB: PMB has a relative confined conformation for LPS binding via both charge and hydrophobic interactions. However, the binding affinity of PMB and LPS is only about a few μM range. In comparison, we can freely engineer the charge and hydrophobic moieties in TD, which promise the identification of stronger LPS binder. As shown in our data in **Figure 1** and **Figure 3**, much efficient LPS binding was observed for both TD nanoparticle and PEGA-TD nanotrap resin than PMB and PEGA-PMB resin, respectively. In addition, PMB is not sufficient for protein binding, instead, TD nanotrap could effectively trap various biomolecules possessing both charge and hydrophobic features.

Secondly, we would like to compare the difference between our TD design and conventional polymers: (a) Dendritic domain of TD has a well-defined structure with multivalent charge and hydrophobic moieties within a close proximity, which foster synergistic effects between charge and hydrophobic moieties in interacting with protein surfaces by forming “salt bridges” in a reduced polarity, similar to the salt bridge formed in the transmembrane domain of membrane

protein. In comparison, the linear polymers with the random charge and hydrophobic moieties on the main chain reduces the synergistic effect due to the spatial restriction. The block-copolymers with the segregated charged and hydrophobic segments further reduce the synergistic effects in protein interaction, due to the separation of two segments. In contrast, TD has “octopus like” flexible scaffold with the spatially adjacent charge and hydrophobic moieties, maximizing the synergy. (b) In addition, block copolymers or random copolymers with charged and hydrophobic moieties generally form micelle or random aggregates, which are not available for protein coating, rather than nonspecific protein adhesion on the surface of nanoparticle. In contrast, TD nanoconstructs forms micelles with CMCs about a few μM , due to the charge repulsion and charge hydrophilicity, which is readily to reassemble with protein with the reverse charges into nanocomplex with much stronger binding affinity of 20-30 nM,^{5,6} therefore, the protein coating by TD nanotrapp in situ is a thermodynamically favorable process.

Since no typical polymer that can represent for all to compare with TD nanotrapp, we instead synthesized a series of TD constructs on PEGA resin to compare the structure-property relationship in protein adsorption. First, we designed a G0 construct on PEGA resin with the random distribution of positively charged amine group and hydrophobic C17, which mimic the random block copolymers with both charge and hydrophobic groups; As a comparison, the same number of amine and C17 were introduced onto PEGA via a lysine branch to generate G1 nanotrapp resin to test the importance of adjacent combination on the synergy in protein binding. Further, we made TD with different branches in PEGA resin, e.g. G2 and G3 as shown in Figure below to compare the valency of TD in protein adsorption, for example, FITC labeled α -LA. As shown in the adsorption kinetics curve below, G1 with close adjacent charge and hydrophobic group showed significant faster and more efficient protein adsorption than G0 with random function distribution. As expected, the protein adsorption was further improved by the increasing valency of TD (unpublished data).

Point by point response:

- (1) this complex architecture which may limit translation into a clinical setting, compared to other simpler and easily assembled polymeric architectures with similar multivalency in charge spreading.

Response: The dendritic TD structure is synthesized by peptide chemistry with high efficiency and precise structure, which is favored for clinical translation in terms of reproducibility and well-defined structure property relationship. As discussed above, conventional polymers are not effective for capturing multiple biomacromolecules in solution, and the undefined structures and batch-to-batch variation largely restrict their clinical application for therapeutic development.

- (2) the necessity to have a hydrophilic tail in the design mode of telodendrimers.

Response: Hydrophilic PEG tail is needed to stabilize the dendritic segment from precipitation.

- (3) Is the self-assembly of such architectures with a CMC of 1-2 micro molar of significance, as it is ultimately forming a complex with LPS for adsorption purposes? Are the nanoassembled structures are first formed which subsequently open and reassemble upon interaction with LPS?

Response: As discussed above, TD with CMC of 1-2 μM is favored for dynamic reassemble with LPS or protein or cell free polynucleotides to form even stronger complex with affinity at low nM ranges.

- (4) For maximum binding in TD-LPS nanocomplex, and its subsequent stability clearly suggests that the dendritic arms need to be extended (also evident from the data provided), so what is the advantage of the telodendrimer architecture and its self-assembly? There is no detailed evaluation of the "rational selection of amphiphilicity" and its influence in tuning the efficacy.

Response: Thank you for your suggestion, the LPS binding affinity can be further optimized as you mentioned to further extend the dendritic arms. In this study, we were aiming to target a much broader range of biomolecules, therefore, the optimal LPS-binding TD may not be the overall optimal construct for attenuating broad range of inflammatory mediators for hyperinflammation attenuation. Therefore, we compared a few representative hydrophobic building blocks, e.g. fatty acid, cholesterol, VE, with charge moieties in TD constructs. As result, C17 was observed to be more effective in protein adsorption, and for general understanding that the flexible C17 has more conformational possibility to fit various surface and pocket for efficient biomolecule binding, therefore was applied for future study. Further optimization is warranted in the future studies.

- (5) What is the immobilization efficiency of PMB on the resin and how does it compare with the telodendrimer nanotrapp? This will play a role and help address the issue of superiority of the telodendrimer nanotrapp in LPS removal. Diffusion characteristics in PMB and telodendrimer immobilized resins are expected to be different on the overall structure of the adsorbed species. LPS removal is more efficient with the telodendrimer model but cytokine removal efficiencies were only somewhat improved from PMB based resin. This is again related to overall structural differences between the two.

Response: PEGA amine resin was converted into carboxylic acid via succinic anhydride, then PMB was conjugated onto PEGA-COOH resin via amide bond formation, which is highly efficient and no comprehensive characterization was conducted. The conjugation of PMB or TD nanotrapp onto

hydrogel resin will influence the swelling properties of PEGA resin, depends on the density of PMB or TD introduced. The PMB density on resin was controlled the same as TD nanotrap, and we compared the LPS adsorption on different resin with the same loading capacity. As shown in Figure 4C in resubmission, PEGA TD resin had a significantly higher efficiency in protein adsorption than PMB-PEGA resin, i.e. 96% vs 74%.

(6) The study makes a claim of “fine-tuned all in one hemoperfusion”, but without listing structure-property relationships of this. The criticism for commercially available Toraymyxin and Cytosorb is misplaced as the claims made here have not been tested clinically and may lead to similar unfortunate outcomes.

Response: Thank you for your comment, we have modified our statement. It was reported that the clinical trials for the LPS-specific Toraymyxin and nonspecific Cytosorb both failed in improving survival of sepsis. Our TD nanotrap approach with different molecular mechanism to target much broader range of inflammatory mediators is promising to improve sepsis treatment, given the significant survival benefits shown in the most clinically relevant CLP mouse sepsis.

1. Bone, R.C., Fisher, C.J., Jr., Clemmer, T.P., Slotman, G.J., Metz, C.A. & Balk, R.A. A controlled clinical trial of high-dose methylprednisolone in the treatment of severe sepsis and septic shock. *N Engl J Med* **317**, 653-658 (1987).
2. Gibbison, B., López-López, J.A., Higgins, J.P.T., Miller, T., Angelini, G.D., Lightman, S.L. & Annane, D. Corticosteroids in septic shock: a systematic review and network meta-analysis. *Critical Care* **21**, 78 (2017).
3. Annane, D., Bellissant, E., Bollaert, P.E., Briegel, J., Keh, D. & Kupfer, Y. Corticosteroids for treating sepsis. *Cochrane Database Syst Rev*, Cd002243 (2015).
4. Fitzgerald, W., Freeman, M.L., Lederman, M.M., Vasilieva, E., Romero, R. & Margolis, L. A System of Cytokines Encapsulated in ExtraCellular Vesicles. *Scientific Reports* **8**, 8973 (2018).
5. Wang, L., Shi, C., Wang, X., Guo, D., Duncan, T.M. & Luo, J. Zwitterionic Janus Dendrimer with distinct functional disparity for enhanced protein delivery. *Biomaterials* **215**, 119233 (2019).
6. Wang, X., Shi, C., Zhang, L., Bodman, A., Guo, D., Wang, L., Hall, W.A., Wilkens, S. & Luo, J. Affinity-controlled protein encapsulation into sub-30 nm telodendrimer nanocarriers by multivalent and synergistic interactions. *Biomaterials* **101**, 258-271 (2016).

Reviewers' Comments:

Reviewer #1:

Remarks to the Author:

In the revised manuscript, the authors have put tremendous effort including new set of animal studies demonstrating efficacy in murine model. Such broad spectrum PAMP/DAMP molecules and proinflammatory cytokines, together with antibiotic treatment, might potentially be effective in sepsis managing. The modulation of properties of resin based on charge and hydrophobicity is extensive and innovative to identify optimal materials with respect to efficacy and side-effect. Nevertheless, the key issues raised by first round of reviewers on viability of this approach for clinical translation and head-to-head comparison to commercial resins in terms of improving performance potentially in the clinic largely remain. In particular, the envisioned clinical use (which should have been more clearly stated in the manuscript) seems based on hemoperfusion but the animal model was based on injection resins to abdominal cavity, creating additional uncertainty for clinical translation.

Reviewer #2:

Remarks to the Author:

The authors have made some effort to assess the therapeutic potential of this nanotrap-based hemoperfusion technique in an animal model of sepsis induced by cecal ligation and puncture. However, with such a small sample size ($n = 8$ animal/group), this experiment should be repeated at least once to ensure reproducibility.

1) It is rather surprising to see a complete protection from the NT(+)/IMI treatment (Figure 6B), and will be important to test its reproducibility by repeat this experiment at least one more time.

2) The survival rate of the IMI group (Figure 6B) did not match with the value predicted from the sample size ($n = 8$), so some explanation would be needed.

Reviewer #3:

Remarks to the Author:

I am convinced through a careful read of the manuscript and the response to concerns that were raised earlier, that the reviewers took the constructive criticism seriously, and made an honest attempt to either address these or provide more evidence that was desired. I am happy to now recommend this manuscript for publication without any further changes.

Point-by-point Response

Reviewer #1 (Remarks to the Author):

In the revised manuscript, the authors have put tremendous effort including new set of animal studies demonstrating efficacy in murine model. Such broad spectrum PAMP/DAMP molecules and proinflammatory cytokines, together with antibiotic treatment, might potentially be effective in sepsis managing. The modulation of properties of resin based on charge and hydrophobicity is extensive and innovative to identify optimal materials with respect to efficacy and side-effect. Nevertheless, the key issues raised by first round of reviewers on viability of this approach for clinical translation and head-to-head comparison to commercial resins in terms of improving performance potentially in the clinic largely remain. In particular, the envisioned clinical use (which should have been more clearly stated in the manuscript) seems based on hemoperfusion but the animal model was based on injection resins to abdominal cavity, creating additional uncertainty for clinical translation.

Response: Thank you for your recognition on the innovation and significance of our reported study. We understand your concern for potential clinical translation as the current form for i.p. injection of resin slurry, which was due to the small blood volume in mice limiting the feasibility for hemoperfusion test. However, we believe that it is straightforward to incorporate these resins into medical devices, for example, patches for local pathogenic molecule adsorption and therapeutic release or cartridges for hemoperfusion use. We have packed mini-column for hemoperfusion test in rats, which will be reported in the future. We will compare the hemoperfusion efficacy compared with the resins tested in the clinical trials for sepsis treatment, although not FDA approved, if it can be made available, due to the potential conflict of interest from the resin/device company. We are confident that our nanotrap resin is readily for clinical testing based on the safety, hemocompatibility after tested in large animals for hemoperfusion treatments. As suggested by the reviewer, we modify the discussion at the end of conclusion section into “TD NT resins are readily to be incorporated into the standard clinical care for sepsis treatment, e.g. local immune intervention or hemoperfusion therapy. If tested effective, it also can be applied to treat other critical illness with high risk of cytokine storm, for example trauma, burn and severe cardiac surgery, etc.”

Reviewer #2 (Remarks to the Author):

The authors have made some effort to assess the therapeutic potential of this nanotrap-based hemoperfusion technique in an animal model of sepsis induced by cecal ligation and puncture. However, with such a small sample size (n = 8 animal/group), this experiment should be repeated at least once to ensure reproducibility.

Response: Appreciate for your recognition. Concern of sample size is addressed as following.

1) It is rather surprising to see a complete protection from the NT(+)/IMI treatment (Figure 6B),

and will be important to test its reproducibility by repeat this experiment at least one more time.

Response: Actually, we have conducted treatments of NT(+)/IMI in three separated experiments on total of 23 severe septic mice with repeated 100% survival (Fig. 6B, Fig S14 and Fig. S15). As evidenced in Figure 5, 7 & 8, NT with optimal charges and treatment schedule can effectively attenuate the excessive inflammation, which is known to cause the progression of sepsis to MODS and mortality. The combination of NT(+) with IMI antibiotics controlling both inflammation and infection is expected to save life, especially with the optimized treatment schedule.

2) The survival rate of the IMI group (Figure 6B) did not match with the value predicted from the sample size ($n = 8$), so some explanation would be needed.

Response: Thank you for your note. This group had seven animals due to one unexpected death during anesthesia. However, significant difference was still observed between NT(+)/IMI group and IMI group with this reduced sample size in statistical analysis. Clarification was added in the legend of Figure 6: (n=8, except for IMI group n=7)

Reviewer #3 (Remarks to the Author):

I am convinced through a careful read of the manuscript and the response to concerns that were raised earlier, that the reviewers took the constructive criticism seriously, and made an honest attempt to either address these or provide more evidence that was desired. I am happy to now recommend this manuscript for publication without any further changes.

Response: We really appreciate for your recognition and comments.

Reviewers' Comments:

Reviewer #2:

Remarks to the Author:

The authors have fully addressed my major concerns.

Point-by-point response to editorial request

NCOMMS-18-08730B

"A Nanotrap Improves Survival in Severe Sepsis by Attenuating Hyperinflammation"

REVIEWERS' COMMENTS:

Reviewer #2 (Remarks to the Author):

The authors have fully addressed my major concerns.

Response: We appreciate that the reviewers agreed that we have fully addressed all their concerns.